# Trajectory Diffusion for ObjectGoal Navigation

**Xinyao Yu**[1,2*], **Sixian Zhang**[1,2*] **Xinhang Song**[1,2], **Xiaorong Qin**[1,2], **Shuqiang Jiang**[1,2,3]

[1]Key Lab of Intelligent Information Processing Laboratory of the Chinese Academy of Sciences (CAS),
Institute of Computing Technology, Beijing, [2]University of Chinese Academy of Sciences, Beijing
[3] Institute of Intelligent Computing Technology, Suzhou, CAS
{xinyao.yu, sixian.zhang, xinhang.song, xiaorong.qin}@vipl.ict.ac.cn
sqjiang@ict.ac.cn

## Abstract

Object goal navigation requires an agent to navigate to a specified object in an unseen environment based on visual observations and user-specified goals. Human decision-making in navigation is sequential, planning a most likely sequence of actions toward the goal. However, existing ObjectNav methods, both end-to-end learning methods and modular methods, rely on single-step planning. They output the next action based on the current model input, which easily overlooks temporal consistency and leads to myopic planning. To this end, we aim to learn sequence planning for ObjectNav. Specifically, we propose trajectory diffusion to learn the distribution of trajectory sequences conditioned on the current observation and the goal. We utilize DDPM and automatically collected optimal trajectory segments to train the trajectory diffusion. Once the trajectory diffusion model is trained, it can generate a temporally coherent sequence of future trajectory for agent based on its current observations. Experimental results on the Gibson and MP3D datasets demonstrate that the generated trajectories effectively guide the agent, resulting in more accurate and efficient navigation. The code is available at https://github.com/sx-zhang/T-diff.git.

## 1 Introduction

Embodied AI aims to develop agents with a comprehensive understanding of their environment, capable of interacting with humans, other agents and entities in real physical environments. As a fundamental task of embodied AI, visual object goal navigation (ObjectNav) task involves placing an agent in an unseen environment and tasking it to navigate to a user-specified object (e.g., 'find a toilet') based on visual sensory input. To efficiently complete the ObjectNav task, the agent needs to construct an information-rich memory to store **past** experiences (i.e., what it has previously seen) to avoid redundant searching. Additionally, it needs to learn a planner to plan the **future** (i.e., determine the most efficient sequence of actions to navigate to the target) to avoid unnecessary exploration.

Humans are innately smart navigators, as they encode both short-term memories from current navigation and long-term memories from daily life into a cognitive map. This map records detailed information about the semantics, positions, and relationships within spatial environments [40]. In practice, the human decision-making process is understood as finding the most possible sequence of releases based on cognitive information [41]. Human planning is a sequential process that ensures temporal consistency and global optimality. However, prior methods of ObjectNav employ single-step planning rather than sequential planning for navigation. As illustrated in Fig. 1, prior end-to-end learning methods [66, 45, 62, 28, 18, 33, 34] formulate their planners as end-to-end networks and train them using reinforcement learning (RL) [66, 45, 62, 28, 18] or imitation learning (IL)

---

*These authors contributed equally to this work.

38th Conference on Neural Information Processing Systems (NeurIPS 2024).

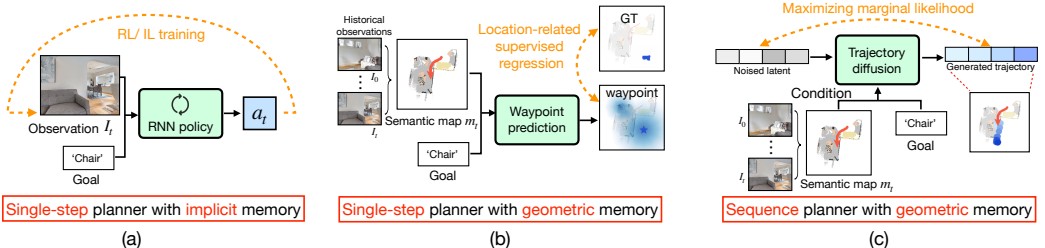

Figure 1: Different planners (denoted as green modules) in existing methods of ObjectNav. (a) represents end-to-end learning methods, and (b) refers to modular methods. (c) denotes the proposed trajectory diffusion, which plans future sequence trajectory based on geometric semantic map.

[33, 34]. These planners perform single-step planning at each moment, outputting an action based on the current egocentric view, which leads to a lack of temporal consistency and interpretability. Furthermore, these methods implicitly encode all past observations, leading to a loss of geometric spatial information, thus limiting their generalization in complex environments [28, 63]. Modular methods [4, 31, 65, 59, 64] attempt to address this by constructing a semantic map during navigation to geometrically store historical observations, helping the agent avoid redundant searching. However, their decision-making process is still single-step. They train a waypoint predictor via supervised learning to plan the next subgoal based on the current semantic map and target. Motivated by learning sequential planning, we propose to model the conditional distribution of the trajectory sequence, i.e., learning to plan a sequence of future trajectory conditioned on the current semantic map and target.

Since both the semantic map and trajectory sequence are high-dimensional, the feature space and computational complexity are significantly high. Thus, directly learning this conditional distribution is challenging [1]. Recently, diffusion models have achieved great success in expressing complex distributions [16, 54] and generating high-quality images [37, 35, 32]. Diffusion models gradually add noise to the data during the diffusion process, transforming the data distribution into a simple distribution (e.g., Gaussian distribution). In the reserve process (i.e., denoising process), noise is iteratively predicted and removed, generating complex data distributions from the simple random distribution. This iterative refinement learning manner enhances the stability and controllability of learning high-dimensional data distributions. Therefore, we utilize diffusion models to learn this conditional distribution and propose the trajectory diffusion (T-diff) model as a navigation planner, which performs sequence planning and generates the future trajectory sequence for the agent.

In this paper, we propose the trajectory diffusion for the ObjectNav task. The trajectory diffusion is a sequence planner designed to generate optimal trajectory sequence for an agent based on its historical observations (i.e., semantic maps) and target object. Specifically, we collect optimal trajectories by using precise maps in training rooms. Then, an agent equipped with a semantic map module is driven to follow these trajectories, gathering semantic maps and poses at each timestep along the way. The collected data are further divided into data pairs consisting of a semantic map at a given moment and the corresponding future trajectory segments. Based on the collected data pairs, we employ DDPM [16] to train our trajectory diffusion. Our trajectory diffusion is implemented by modifying Transformer-based diffusion, which takes noised latent as the starting input and iteratively refines them to produce trajectory sequence. Once the trajectory is generated, we drive the agent to move along the predicted trajectory sequence until the target is found. Evaluations on the Gibson [47] and MP3D [3] simulators show our trajectory diffusion model significantly outperforms baselines. Visualization results confirm effective guidance from generated trajectories. Additionally, we further showcase the scalability of our trajectory diffusion model across different simulators.

## 2   Related Work

**ObjectGoal Navigation.** Goal-driven navigation [22, 42, 43, 44, 63] is a fundamental task in embodied AI, and this paper focuses on a specific variant of this task, namely the ObjectNav task, where the goal is defined by object semantics. Current works for ObjectNav task can be categorized into two types: end-to-end learning methods and modular methods. End-to-end learning methods develop a navigation policy by interacting with the environment trained by reinforcement learning (RL) [45, 66, 61] or imitation learning (IL) [34, 33]. These approaches typically take inputs such as

target object categories and extra information, including visual representations[20, 29, 18] and object relationship graphs[51, 62, 11, 55]. Then they predict single-step planning in each timestep. The end-to-end method implicitly encodes all past observations, which results in the loss of geographic spatial information. Consequently, its generalization ability in environments with complex layouts is limited. Modular methods [4, 5, 6] typically preserve a top-down geometric map enriched with semantic details. They are also single-step planners that predict waypoints as sub-goals at each time step based on the constructed semantic map. Through supervised learning, PONI[31] learns to predict the nearest frontier with the current semantic map to infer where to explore, while PEANUT[59] directly predicts the target location in the form of a probabilistic goal map. Current end-to-end learning methods are single-step planners with implicit memory, while modular methods are single-step planners with geometric memory. Alternatively, our trajectory diffusion approach is a sequence planner with geometric memory, predicting future sequential trajectories based on the current semantic map. Sequence planning ensures temporal consistency and interpretability of decisions. The geometric memory prevents the agent from redundant exploration.

**Diffusion Model.** Diffusion models (DDPMs [16]) are generative models that learn complex data distributions by iteratively predicting and removing randomly sampled noise to obtain target samples. Diffusion models have been successfully applied in image-related fields, including image generation[32, 37, 30], super-resolution[36, 46], image inpainting[26, 48], and image editing[2, 39]. Diffusion models predominantly use UNet-based [36, 37] and Transformer-based [30, 32, 46] architectures. UNet excels in preserving spatial details for high-resolution images, while Transformers capture global context, suited for sequential data. Therefore, we adopt a Transformer-based diffusion model to implement our trajectory diffusion. Recently, diffusion models are gradually employed in the field of robotics. several works [8, 58, 25, 13] leverage Data collection from real-world often suffers from scarcity or lacks diversity. As natural data synthesizers, diffusion models are leveraged by several works [8, 58, 25, 13] for data augmentation purposes. Furthermore, methods like Diffuser[17], Crossway Diffusion[21] and Diffusion policy [9] utilize diffusion models to fit multi-modal behavioral data of agents. In line with our work, the proposed Diffusion Trajectory aims to learn the distribution of trajectory sequence conditioned on semantic maps and the goal, primarily to address the ObjectNav task.

## 3 Preliminaries of ObjectNav

The task of ObjectNav involves navigating an agent to a specific type of object (e.g., 'chair') in unseen environments. At the beginning of each episode, the agent is initialized at a random position. During navigation, at every timestep $t$, the agent receives egocentric RGB-D images $I_t$, the target object $o$, and the senor pose $p_t$, which includes the spatial coordinates and the direction the agent is facing. The agent performs one of several discrete actions, includeing `move_forward`, `turn_left`, `turn_right`, and `stop`. The action `stop` is autonomously initiated by the agent once it determines to complete the task. An episode is considered successful if the agent stops within a preset number of steps at a spot where the target object is within a specified distance (e.g., less than $1m$) and is visible in the agent's field of view.

Existing works for the ObjectNav task can be categorized into end-to-end learning methods and modular methods, as shown in Fig. 1. The end-to-end learning methods [66, 18, 33, 34] generate single-step plans at each time step, formulated as a policy $\pi(a_t|I_t, o)$, where $a_t$ denotes the action at timestep $t$. They typically utilize RL or IL to train policy functions $\pi(a_t|I_t, o)$. The training objective of RL is to maximize the expected sum of discounted rewards. Let $\chi$ denote a sequence of object, action, reward tuples sampled based on $\pi$ within an episode. The training objective is $\underset{\pi}{argmax}\mathbb{E}_{\chi\sim\pi}\left[R_\chi\right], R_\chi = \sum_{t=1}\gamma^{t-1}r_t$, where $\gamma$ is a discount factor, and $r_t$ denotes the reward function, typically implemented as a sparse success reward. This is because dense rewards can inhibit agent's exploration [28], thus impairing its generalization in unseen environments [33]. Sparse rewards are desirable, but they result in most sample trajectories having difficulty obtaining positive rewards, making the learning process challenging. Moreover, as the policy parameters are updated, previously sampled trajectories become obsolete, necessitating the collection of new data. Consequently, the low sample efficiency of end-to-end learning methods results in high computational costs for training. As for IL-based methods [33, 34], the training objective is to minimize the difference between the policy's output and human demonstrations (i.e., behavior cloning), summarized as $\underset{\pi}{argmin}\mathbb{E}_{(I_t^d, a_t^d)\sim\mathcal{D}}[-\log(\pi(a_t^d|I_t^d, o))]$, where $\mathcal{D}$ is a dataset of human

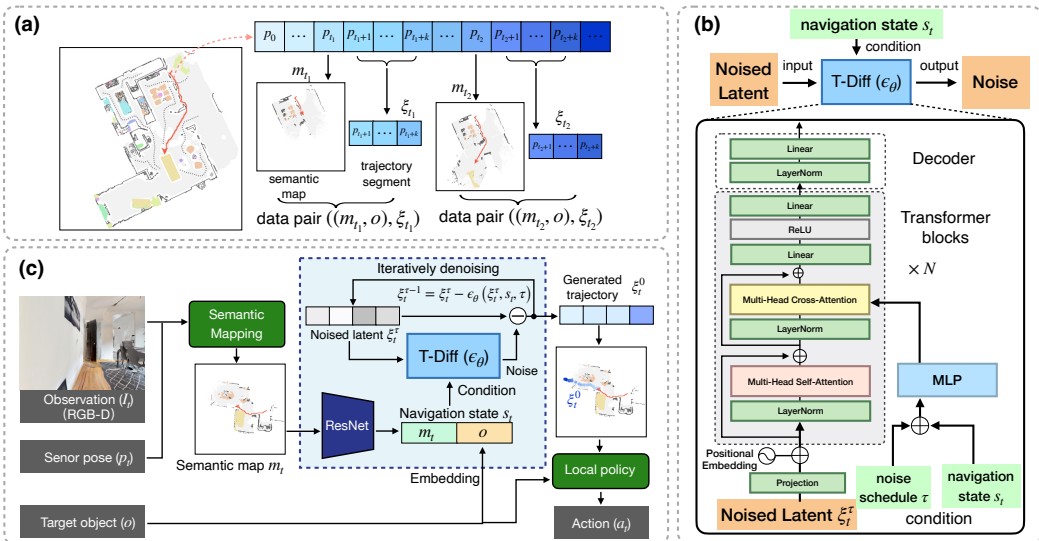

Figure 2: Frameworks of the trajectory diffusion (T-Diff). (a) refers to the process of dividing the collected data into data pairs. (b) shows the implementation structure of T-Diff. (c) illustrates the navigation process guided by trajectories generated by T-Diff.

demonstrations. However, collecting human demonstrations is highly expensive, making IL training costly. Additionally, end-to-end learning methods implicitly encode historical observations, which results in a lack of geometric memory. This also limits the generalization capability of these methods.

Modular methods [4, 31, 65, 59] typically construct a local semantic map (the map module will be detailed in Sec. 4.2) during navigation. The semantic map geometrically stores historical observations, helping the agent avoid redundant exploration. For the navigation planner, they formulate it as $f(\Omega|m_t, o)$, where $\Omega$ is a set of points in $m_t$, e.g., $\Omega$ can be defined as frontiers of $m_t$ [31] or unknown regions outside $m_t$ [59]. The module $f(\Omega|m_t, o)$ is trained as a supervised regression task with the training loss $\sum_{\Omega} \mathcal{L}\left(V\left(\Omega\right), f\left(\Omega|m_t, o\right)\right)$, where $V\left(\Omega\right)$ represents the ground truth, and its values are calculated based on the actual location of the target (i.e. the closer $\Omega$ is to the target, the higher value of $V(\Omega)$). Based on the predictions of $f$, the agent selects the point with the optimal value as a sub-goal to guide the agent. The planner $f$ re-predicts at each timestep, which is also a form of single-step planning. Supervised learning is efficient for training, however, due to the limited room diversity (i.e., current simulators [47, 3, 49] mostly contain fewer than 1000 unique rooms), these location-related supervision constraints lead the learned planner $f$ to overfit to the layouts of the training rooms [63].

In summary, end-to-end learning methods learn a **single-step planner with implicit memory**, and their learning process suffers from sample inefficiency and high training costs. On the other hand, modular methods learn a **single-step planner with geometric memory**, but the generalization of their planner is constrained by location-related supervision. In our work, our trajectory diffusion model is a **sequence planner with geometric memory**, which aims to learn the conditional distribution $p(\tau_t|m_t, o)$, where $\tau_t$ represents the planned future sequential trajectory. The sequential plan ensures temporal consistency and interpretability of decisions, while the geometric memory ensures efficient exploration. Furthermore, in terms of model training, we leverage DDPM with automatically collected trajectories to learn this conditional distribution, effectively avoiding sample inefficiency, the necessity for expensive human demonstration collection, and the risk of overfitting location-related information.

## 4 Trajectory Diffusion

### 4.1 Diffusion Model

Diffusion models are probabilistic generative models trained to learn data distributions by iteratively denoising variables sampled from Gaussian distributions. The forward process of diffusion models (DDPMs [16]) is defined as the diffusion process, which is implemented via a Markov chain that

gradually applies Gaussian noise to the real data which can be formulated as follows:

$$q\left(x_{1:T}|x_0\right) = \prod_{\tau=1}^{T} q\left(x_\tau|x_{\tau-1}\right), \; q\left(x_\tau|x_{\tau-1}\right) = \mathcal{N}\left(x_\tau; \sqrt{\bar{\alpha}_\tau}, \left(1 - \bar{\alpha}_\tau\right)\mathbf{I}\right) \tag{1}$$

where constants $\bar{\alpha}_\tau$ are hyper-parameters. $x_0$ is the real data, while $x_1, \ldots, x_T$ are noised latent data. The dimension of both latent and real data are the same, i.e., $x_{0:T} \in \mathbb{R}^d$. By applying the reparameterization trick, the noised data can be sampled by $x_\tau = \sqrt{\bar{\alpha}_\tau}x_0 + \sqrt{1 - \bar{\alpha}_\tau}\epsilon_\tau$, where $\epsilon_\tau \sim \mathcal{N}(0, \mathbf{I})$. Diffusion models learn the real data distribution by reversing the diffusion Markov chain, denoted as $p(x_{\tau-1}|x_\tau)$, which is referred to as denoising process. Theoretically, this process reduces to predict the noise added to $x_\tau$. The noise prediction network $\epsilon_\theta$ is trained using the mean-squared error between the predicted noise and the ground truth sampled Gaussian noise $\epsilon_\tau$. Additionally, diffusion models can be conditioned on other inputs (e.g., text [2, 30] or image [9, 52]), then the training objective is formulated by

$$\mathcal{L}_\theta = \mathbb{E}_{x,c,\epsilon,\tau}[\|\epsilon_\tau - \epsilon_\theta\left(x_\tau, c, \tau\right)\|_2^2] \tag{2}$$

where $c$ is the embeddings of input condition. Once the model $\epsilon_\theta$ is trained, real data are iteratively generated starting from random noise.

## 4.2 Navigating with Trajectory Diffusion

Our trajectory diffusion aims to generate the optimal future trajectory for the ObjectNav agent based on its current state, thereby assisting the agent in efficiently navigating to the target object.

**Trajectory collection.** To train the trajectory diffusion, we collect a set of data pairs $((m_t, o), \xi_t)$, where $m_t$ is the semantic map at time $t$, $o$ denotes target object, and $\xi_t \in \mathbb{R}^{2 \times k} = [p_{t+1}, \ldots, p_{t+k}]$ represents the trajectory segment, defined as the concatenation of the senor poses from time $t + 1$ to $t + k$. Each sensor pose $p_t$ is recorded in 2 dimensions representing the 2D coordinates. To obtain efficient trajectories and corresponding semantic maps, we first randomly initialize a start position in the training room. Then, the Fast Marching Method (FMM) [38] is utilized to compute an optimal path from the start position to a specific target $o$ based on the precise collision maps of the current training room. Note that such precise maps are unavailable during testing since the test rooms are unseen. Subsequently, we drive an agent equipped with semantic mapping module along this optimal path, recording the semantic map $m_t$ at each timestep. Based on the collected data, we sample a series of data pairs for training our trajectory diffusion. Furthermore, the collected trajectories are segmented into sub-trajectories $\xi_t$ of length $k$. Together with the corresponding semantic maps $m_t$ and target object $o$, these data pairs $((m_t, o), \xi_t)$ ultimately constitute the training data for our trajectory diffusion, as illustrated in Fig. 2 (a).

**Trajectory diffusion model.** We utilize DDPM to train our trajectory diffusion to estimate the conditional distribution $p(\xi_t|m_t, o)$. In the diffusion process, we sample noised data by $\xi_t^\tau = \sqrt{\bar{\alpha}_\tau}\xi_t^0 + \sqrt{1 - \bar{\alpha}_\tau}\epsilon_\tau$, where $\epsilon_\tau \sim \mathcal{N}(0, \mathbf{I})$ represents Gaussian noise and $\xi_t^0$ is the real data $\xi_t$, i.e., the trajectory segments in the collected data pairs $((m_t, o), \xi_t)$. $\bar{\alpha}_\tau$ and $\tau$ are hyper-parameters that control the variance schedule. Note that two timesteps ($t$ and $\tau$) are involved here, where $t$ represents a specific moment in navigation, and $\tau$ denotes the noise schedule in diffusion or denoising processes. We train the model to predict the added noise, and the training objective is modified from Eq. 2

$$\mathcal{L}_\theta = \mathbb{E}_{\xi,s,\epsilon,\tau,t}[\|\epsilon_\tau - \epsilon_\theta\left(\xi_t^\tau, s_t, \tau\right)\|_2^2] \tag{3}$$

where $\theta$ represents the parameter for trajectory diffusion. The navigation state $s_t$ of the current agent acts as the condition for trajectory diffusion. The state $s_t$ is the concatenation of the embedding of the semantic map $m_t$ and the target object $o$. The semantic map $m_t$ is encoded by a ResNet18 (without pretrained), while the target object $o$ is encoded using linear projection.

Regarding the implementation of trajectory diffusion, since the navigation trajectory $\xi_t$ is a sequence of $k$ tokens, each with a dimension of $m$, our trajectory diffusion follows DiT [30], which is a diffusion model based on the Transformer architecture. As illustrated in Fig. 2 (b), the noised latent $\xi_t^\tau$ is encoded via a linear projection with added positional embeddings, and then processed through a series of transformer blocks. In addition to the noised latent input, trajectory diffusion also conditions on the diffusion timestep $\tau$ and the navigation state $s_t$. The condition information interacts with the encoded noised latent through a multi-head cross-attention layer. After passing through $N$

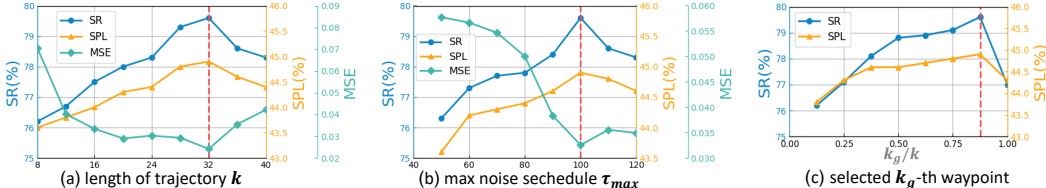

Figure 3: The impact of three hyper-parameters. (a) represents the impact of the length of generated trajectories $k$ during training. (b) reflects the impact of maximum noise schedule $\tau_{max}$ in diffusion and denoising process of T-diff. (c) shows the impact of selected proportion of generated trajectory length for navigation performance.

Transformer blocks, we utilize a standard linear decoder to output the noise prediction. Note that the output has a shape equal to the original noised latent.

**Navigating with generated trajectory.** During navigation, at each timestep $t$, the agent receives the current egocentric RGB-D view $I_t$, the current sensor pose $p_t$, and the target object $o$ as shown in Fig. 2 (c). The **semantic mapping module** aids the agent in constructing an incrementally growing semantic map $m_t$. At each timestep during navigation, the received RGB-D image is segmented into semantic categories using a pre-trained segmentation model [14]. Subsequently, the depth image is employed to project each pixel, along with its semantic label, into 3D space. Points within a specific height range (relative to the agent's height) are designated as obstacles. Then, This point cloud is transformed into a voxel occupancy grid, which is integrated across the height dimension to create an egocentric map. The egocentric map is then transformed into an allocentric coordinate system based on the agent's pose and is aggregated with the pre-existing global map. Consequently, the semantic map $m_t \in \mathbb{R}^{(4+n) \times h \times w}$ comprises $(n + 4) \times h \times w$ elements, where $n$ denotes the number of semantic categories, and $h$, $w$ are the map size. Channels 1 and 2 depict the obstacle map and the explored regions, respectively. Channels 3 and 4 represent the agent's current position and all previous positions.

The embeddings of the semantic map $m_t$ and target object $o$ are concatenated to form the navigation state $s_t$, serving as generation condition. The trained trajectory diffusion model $\epsilon_\theta$ begins with an initialized Gaussian noise $\xi_t^{\tau_{max}}$ as the initial input, and predicts the noise $\epsilon_\theta(\xi_t^\tau, s_t, \tau)$ contained within the noised latent $\xi_t^\tau$. Then, the noised latent trajectory is denoised by $\xi_t^{\tau-1} = \xi_t^\tau - \epsilon_\theta(\xi_t^\tau, s_t, \tau)$. This denoising process is repeated for $\tau_{max}$ steps, iteratively generated the final trajectory $\xi_t^0$.

Once the generated trajectory $\xi_t^0$ (i.e., $\xi_t$) is obtained, a local policy [4, 6] is employed to drive the agent along this trajectory. To prevent the agent from encountering unreachable points (i.e., obstacles) in the generated trajectory, we select the $k_g$-th waypoint of $\xi_t$ as the navigation goal, where $k_g$ is a hyper-parameter discussed in Sec. 5.2. The local policy converts the navigation goal into low-level actions by computing a collision-free path using the FFM method based on the obstacle channel from the semantic map $m_t$. It then determines deterministic actions according to the agent's step distance to navigate agent towards the navigation goal (i.e., $k_g$-th waypoint of $\xi_t$). The trajectory diffusion generates a new trajectory every $t_{T-diff}$ step, while the local policy replans deterministic action at each step of navigation.

## 5 Experiments

### 5.1 Experimental Setup

**Dataset.** We evaluate the performance of our model on standard ObjectNav datasets, including Gibson [47] and Matterport3D (MP3D) [3] , in the Habitat simulator. For Gibson, we use 25 train / 5 val scenes from the tiny-split, following the settings of [31], with 1000 validation episodes containing 6 target object categories. For MP3D, we utilize 56 train / 11 val scenes, with 2195 validation episodes containing 21 target object categories. The detailed goal categories are mentioned in Appendix.

**Implementation Details.** For the training of trajectory diffusion model, we sample 84k and 465k data pairs from the training scenes in Gibson and MP3D (85% train / 15% val), respectively. The semantic maps are resized to $224 \times 224$. We implement the trajectory diffusion model based on the DiT [30] structure. Additionally, the semantic map in condition information is encoded by

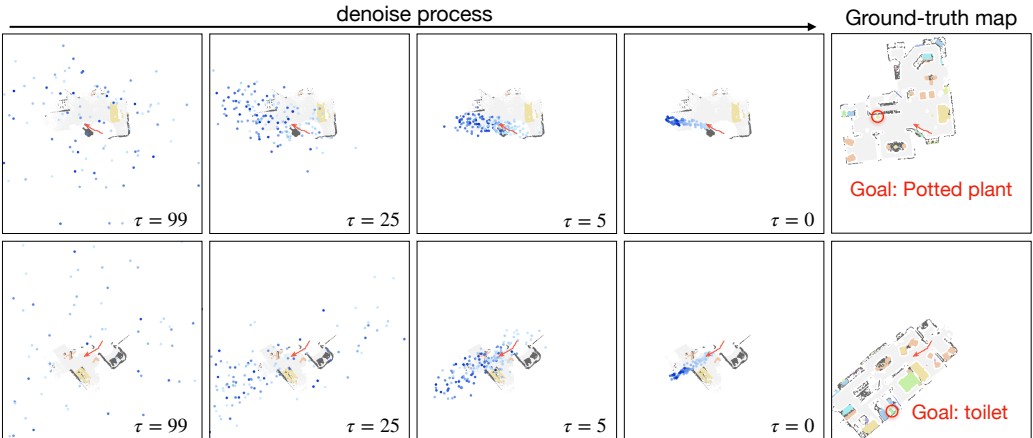

Figure 4: Visualization of trajectory generation. Each row in the figure overlays the results of five repeated experiments, where the same semantic map and target are used as conditions but different random initialization noises are used as inputs. The color intensity of trajectory points represents temporal order, with lighter colors indicating points closer in time to the current navigation timestep.

a ResNet-18 [15] with the first convolutional layer's input channel adjusted to accommodate the dimension of the semantic map. The diffusion process contains $N = 8$ Transformer blocks. Training is performed using AdamW optimizer[19, 24] with a base learning rate of 1e-4, warmed up for 1000 steps using linear warmup and cosine schedule. After the warmup steps, the learning rate for the diffusion model is decayed by a factor of 1e-3, and the learning rate of the semantic map encoder is decayed by a factor of 1e-6. Each model is trained for 200 epochs. Following some common generative methods, exponential moving average(EMA) is used with a maximum decay of 0.9999 during training. All reported results are obtained using EMA model. The maximum noise schedule $\tau_{max}$ is set to 100. The length of the predicted trajectory $k = 32$ and the selected $k_g$-th point is set to 28. The experiments on these three hyper-parameters are detailed in Sec. 5.2. The agent's turn angle is fixed at 30 degrees and each $Forward$ step distance is 25 cm. The maximum timestep limit is set to 500 during navigation and $t_{T-diff}$ is set to 5. Note that, since the navigation performance of multiple repeated experiments does not show significant differences, error bars are not reported.

**Evaluation metrics.** Three standard metrics are utilized to quantify the performance of the model in ObjectNav task, following [31, 59, 4]. **SR** (Success Rate) indicates the proportion of success episodes. **SPL** (Success weighted by Path Length) represents the success rate of episodes weighted by path length, thereby reflecting the efficiency of the agent's path relative to the shortest path. **DTS** (Distane To Goal) denotes the distance of the agent from the goal when the episode ends. In addition, we use **MSE** (Mean Squared Error), which reflects the distance between the denoised generated trajectory and the real trajectory, to assess generation accuracy of our T-Diff.

## 5.2 Evaluation Results

**Hyper-parameter tuning of T-Diff.** Hyper-parameter $k$ determines the length of generated trajectory. We evaluate the impact of parameter $k$ on the generated trajectory (on MSE metric) and navigation performance (on SR and SPL metrics), as shown in Fig. 3 (a). The results indicate that both overly short and overly long generated trajectories are suboptimal. We infer that when the trajectory length is too short, original sequence-planning gradually turns into step-planning, which undermines the temporal consistency of the planning. Conversely, if the trajectory is too long (i.e., greater than 32), the distribution $p(\xi_t|m_t, o)$ becomes more complex, making the learning process more difficult. Based on experimental results, we determine that $k = 32$ is the optimal value.

The hyper-parameter $\tau_{\text{max}}$ represents the maximum noise schedule, determining the upper limit of iterations in both the diffusion and denoising processes. As evaluation results shown in Fig. 3 (b), the noise schedule $\tau_{\text{max}}$ is empirically set to 100.

**Visualization of trajectory generation.** We visualize the iterative trajectory generation process, as shown in Fig. 4. Initially, the trajectories are initialized as random coordinate points. As the denoising process progresses, these points gradually converge, forming the final trajectory at $\tau = 0$. Note that

Table 1: Ablation study on minimal trajectory length for T-Diff training in Gibson (val). When the trajectory length is 1 (rows 2-6), T-Diff is trained to directly predict a waypoint. $i$-$th$ denotes that the training ground truth is the point at the i-th step from the current position on the optimal trajectory.

| ID | Method | Trajectory | | | Navigation (Gibson) | | |
|---|---|---|---|---|---|---|---|
| | | Length | $i$-$th$ | MSE ↓ | SR(%) ↑ | SPL(%) ↑ | DTS(m) ↓ |
| 1 | Single-step (PONI) | - | - | - | 73.6 | 41 | 1.25 |
| 2 | T-Diff | 1 | 1 | 0.2247 | 71.0 | 37.6 | 1.49 |
| 3 | T-Diff | 1 | 8 | 0.2253 | 72.8 | 40.4 | 1.44 |
| 4 | T-Diff | 1 | 16 | 0.2308 | 72.6 | 40.8 | 1.45 |
| 5 | T-Diff | 1 | 24 | 0.2291 | 74.2 | 42.1 | 1.39 |
| 6 | T-Diff | 1 | 32 | 0.2456 | 73.8 | 41.3 | 1.40 |
| 7 | T-Diff | 4 | 1 | 0.1042 | 75.9 | 43.1 | 1.32 |
| 8 | T-Diff | 32 | 1 | 0.0357 | 79.6 | 44.9 | 1.00 |

the visualization is conducted on the validation set of Gibson, where scene layout is unknown to our trajectory diffusion method. Despite this, the trajectory diffusion is still capable of generating the most efficient path to the target, as indicated by the goal position marked on the ground-truth map.

**Ablation study on minimal trajectory length for T-Diff training.** We conduct an ablation study using minimal trajectory lengths (e.g., 1 and 4) for training T-Diff, as shown in Tab. 1. Note that when the length is set to 1, the prediction of T-Diff is a single waypoint. The results indicate that when the length is set to 1, T-Diff's performance is influenced by the choice of ground truth point (i.e., the i-th point from current position on the optimal trajectory). The performance with shorter lengths (1 or 4) is lower compared to longer lengths (32).

We hypothesize that predicting a sequence of trajectories, as opposed to a single point, allows each predicted point to receive contextual information from neighboring points. This helps correct and smooth out prediction errors of individual points, reducing the sensitivity of the results to single-point errors. Consequently, this ensures more stable predictions and enhances overall accuracy of trajectory prediction. This finding further supports our motivation for using sequence planning.

**Evaluations of T-Diff variants.** We compare different variants of T-Diff as shown in Tab. 2, where rows 2-4 represent different variants of T-Diff (i.e., sequence planner with geometric memory). The comparison in row 1 uses an enhanced FBE method (proposed by PONI [31]) combined with a local policy (i.e., single-step planner with geometric memory). The 'X' marks in row 1 indicate that T-Diff is not used, but this alternative

Table 2: Comparison with different variants of T-Diff on Gibson (val). $I_t$ means using RGB information as condition while $m_t$ refers to employing semantic map as condition. Here, LP denotes local policy, and FBE corresponds to the area potential function proposed by [31].

| ID | Method | T-Diff variants | Trajectory | Navigation (Gibson) | | |
|---|---|---|---|---|---|---|
| | | | MSE ↓ | SR(%) ↑ | SPL(%) ↑ | DTS(m) ↓ |
| 0 | Random policy | ✗ | - | 0.4 | 0.4 | 3.89 |
| 1 | FBE+LP | ✗ | - | 72.3 | 38.5 | 1.32 |
| 2 | T-Diff+LP | Visual($I_t$) | 0.2058 | 73.5 | 41.6 | 1.23 |
| 3 | T-Diff+LP | Visual ($m_t$) | 0.0546 | 76.9 | 44.1 | 1.08 |
| 4 | T-Diff+LP | Visual ($m_t$)+Goal | 0.0357 | 79.6 | 44.9 | 1.00 |

still employs semantic map and goal for navigation. Row 0 refers to the case where navigation process does not use any map or goal. The comparison between row 1 and T-Diff variants (rows 2-4) demonstrates that sequence planning achieves superior navigation performance. Moreover, compared to different T-Diff variants, row 2 represents the variant that utilizes only a single timestep observation $I_t$ for trajectory generation. Rows 3 and 4 represent variants that use a semantic map $m_t$ encompassing all historical observations as conditions for trajectory generation. Comparing rows 2 and 3, the results indicate that using $m_t$ not only improves trajectory generation but also enhances navigation performance. We hypothesize that navigation is a sequential decision-making task, and relying solely on current single-step observations can lead to suboptimal decisions due to a lack of temporal consistency, thereby reducing overall navigation efficiency.

Furthermore, comparing rows 3 and 4, the inclusion of target object improves navigation accuracy, demonstrating that the agent's trajectory is goal-driven rather than aimless. Finally, compared to the baseline (comparing rows 1 and 4), integrating our trajectory diffusion method improves navigation performance by 7.3% in SR, 6.4% in SPL, and reduces DTS by 0.32m. These results validate the effectiveness of the proposed trajectory diffusion.

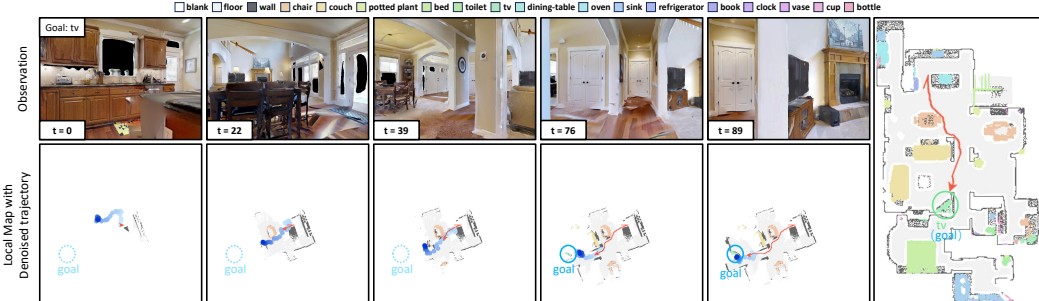

Figure 5: Visualization of navigating with generated trajectory on Gibson (val). The top row shows the agent's first-person RGB observation and the bottom row displays the local semantic map along with generated trajectory by T-Diff. The right figure shows the ground truth map with the actual trajectory of the agent and the target location marked. The generated trajectory effectively guides the agent towards the target, even when the target remains unseen.

**Comparisons with simpler model for trajectory generation.** We consider the following simple decoder to learn the trajectory generation $P(\tau_t|m_t, o)$ as a comparison. It adopts a similar Transformer-based architecture to T-Diff, with comparable parameters and the same conditional

Table 3: Comparisons with simpler model for trajectory generation. MSE measures the quality of generated trajectories, while SR, SPL, and DTS indicate navigation performance.

|  | MSE ↓ | SR(%) ↑ | SPL(%) ↑ | DTS(m) ↓ |
|---|---|---|---|---|
| Simple decoder | 0.6541 | 59.2 | 33.5 | 2.05 |
| T-diff (Ours) | 0.0357 | 79.6 | 44.9 | 1.00 |

inputs. However, unlike T-Diff, which is trained using DDPM, this competitor is trained with MSE loss. The results are shown in the Tab. 3. The results indicate that directly learning $P(\tau_t|m_t, o)$ through supervised training yields poor performance. Our analysis suggests that since both $m_t$ and $\tau_t$ are high-dimensional, the target distribution $P(\tau_t|m_t, o)$ is also high-dimensional. Given the limited number of training rooms (less than 100), $P(\tau_t|m_t, o)$ is sparse and difficult to learn directly. In contrast, the diffusion model (DDPM), through its diffusion and denoising process, gradually simplifies complex distribution into multiple simpler distributions. This allows for better learning of $P(\tau_t|m_t, o)$ distribution. Therefore, our experiments and analysis confirm the necessity of using diffusion models for learning trajectory generation.

**Scalability.** We compare the performance of our trajectory diffusion method with the modular method (i.e., PONI [31]) across different simulators, as shown in Tab. 4. When training and testing are performed on the same simulator, despite the testing rooms being unseen, the layouts of the training and testing rooms are still similar.

Table 4: Comparisons of navigation performance across different training and testing simulators.

| ID | Method | Train | Test | SR(%) ↑ | SPL(%) ↑ | DTS(m) ↓ |
|---|---|---|---|---|---|---|
| 1 | PONI [31] | Gibson | Gibson | 73.6 | 41.0 | 1.25 |
| 2 | PONI [31] | MP3D | Gibson | 43.9 | 26.3 | 2.56 |
| 3 | T-Diff (Ours) | Gibson | Gibson | 79.6 | 44.9 | 1.00 |
| 4 | T-Diff (Ours) | MP3D | Gibson | 78.2 | 45.2 | 1.07 |

similar. In this scenario, the modular method, which uses location-related information for supervision, achieves relatively good performance, as shown in line 1 of Tab. 4. However, when the training and testing rooms come from different simulators, the performance of the modular method significantly deteriorates (line 2). In contrast, our trajectory diffusion method demonstrates better scalability. Even when the training and testing rooms come from different simulators, it maintains good navigation performance (compare lines 3 and 4).

**Navigation with trajectory diffusion.** The generated trajectory $\xi_t$ represents the future location points of the agent from time $t + 1$ to $t + k$. As mentioned in Sec.4.2, instead of directly using the first points on the generated trajectory as the waypoint to guide the agent, we select the $k_g$-th point as the guidance. The impact of the hyper-parameter $k_g$ is evaluated as shown in Fig. 3 (c), where the horizontal axis represents $k_g/k$. The results indicate that the optimal value is achieved when $k_g/k = 0.875$, i.e., when $k = 32$, $k_g = 28$. We attribute this to the fact that, while the generated trajectory follows the correct overall trend, it still fluctuates within a certain range, as shown in Fig. 4. Therefore, if the selected point is too close to the current coordinate, the planning of near-term actions is more frequently affected by these fluctuations, leading to a performance decline.

Table 5: Comparing ObjectNav performance on Gibson and MP3D of related studies. Note that Red-Rabbit [53] utilizes auxiliary tasks for training, while THDA [28] and Habitat-Web [34] incorporate additional training data. Results of SemExp [4], L2M [12] and Stubborn [27] are reported from [60]. Results marked with * indicate our implementation.

| ID | | Method | Gibson | | | MP3D | | |
|---|---|---|---|---|---|---|---|---|
| | | | SR(%) ↑ | SPL(%) ↑ | DTS(m) ↓ | SR(%) ↑ | SPL(%) ↑ | DTS(m) ↓ |
| | 1 | Random | 0.4 | 0.4 | 3.89 | 0.5 | 0.5 | 8.05 |
| I | 2 | DD-PPO [44] | 15.0 | 10.7 | 3.24 | 8.0 | 1.8 | 6.94 |
| | 3 | Red-Rabbit [53] | - | - | - | 34.6 | 7.9 | - |
| | 4 | THDA [28] | - | - | - | 28.4 | 11.0 | 5.62 |
| | 5 | SSCNav* [23] | - | - | - | 27.1 | 11.2 | 5.71 |
| | 6 | EmbCLIP* [18] | 68.1 | 39.5 | 1.15 | 29.2 | 10.1 | 5.40 |
| | 7 | Habitat-Web [34] | - | - | - | 35.4 | 10.2 | - |
| | 8 | ENTL [20] | - | - | - | 17.0 | 5.0 | - |
| | 9 | OVG-Nav [56] | - | - | - | 35.8 | 12.3 | 5.69 |
| II | 10 | FBE [50] | 64.3 | 28.3 | 1.78 | 22.7 | 7.2 | 6.70 |
| | 11 | ANS [6] | 67.1 | 34.9 | 1.66 | 27.3 | 9.2 | 5.80 |
| | 12 | SemExp [4] | 71.1 | 39.6 | 1.39 | 28.3 | 10.9 | 6.06 |
| | 13 | PONI [31] | 73.6 | 41.0 | 1.25 | 31.8 | 12.1 | 5.10 |
| | 14 | L2M [12] | - | - | - | 32.1 | 11.0 | 5.12 |
| | 15 | Stubborn [27] | - | - | - | 31.2 | 13.5 | 5.01 |
| | 16 | 3D-aware [60] | 74.5 | 42.1 | 1.16 | 34.0 | 14.6 | **4.78** |
| | 17 | L3MVN [57] | 76.9 | 38.8 | 1.01 | - | - | - |
| | 18 | SGM [64] | 78.0 | 44.0 | 1.11 | 37.7 | 14.7 | 4.93 |
| | 19 | **T-Diff (Ours)** | **79.6** | **44.9** | **1.00** | **39.6** | **15.2** | 5.16 |

Additionally, we visualize the navigation process of the agent, as shown in Fig. 5. According to the ground truth map and target location on the right side, it is evident that during navigation, the trajectories generated by our trajectory diffusion are consistently correct and efficient, even when the target object is not visible in the early stages (t<76). Consequently, the agent efficiently finds the target guided by the generated trajectories. More visualizations can be found in the Appendix.

**Comparisons with the related works.** We evaluate the performance of our T-Diff on ObjectNav task by comparing it with related baselines, categorizing into end-to-end [44, 53, 28, 23, 18, 34, 20, 56] and modular [50, 5, 4, 31, 12, 27, 60, 57] methods. It is worth noting that some methods incorporate additional information [28, 10, 34] or auxiliary tasks [53, 7], making it challenging to achieve a fair comparison. Therefore, we primarily focus on the following baselines: SemExp [4], PONI [31], OVG-Nav [56], L3MVN [57], L2M [12], and SSCNav [23]. PONI enhances SemExp by introducing supervised learning to predict goal-related information. OVG-Nav uses semantic topological maps to plan sub-goal nodes at a high level for the agent. L3MVN leverages LLM to infer unknown regions based on semantic information of the current boundary. L2M and SSCNav improve navigation by learning to build a top-down map that is egocentric in a single timestep. Since L2M and SSCNav only report results on their custom datasets, we use either re-implementation results from other work [60] with similar experimental settings or our own implementation results for comparison.

We compare our T-Diff with these methods on the validation sets of Gibson and MP3D, as shown in Tab. 5. On the Gibson, our T-Diff outperforms the current state-of-the-art method [57] by 2.7%, 6.1%, and -0.01m in SR, SPL, and DTS metrics, respectively. On the MP3D, T-Diff improves by 3.8%, 2.9%, and -0.53m in the same metrics compared to the current state-of-the-art method [56].

# 6 Conclusion

We propose a trajectory diffusion model (T-Diff) as a sequence planner for ObjectNav task. Our method leverages agent's historical observations and target object as conditions to iteratively generate the future sequential trajectory. Experimental results on standard datasets, including Gibson and MP3D, demonstrate that our T-Diff effectively improves navigation performance compared to the baselines. Furthermore, additional visualizations and experiments, detailed in the Appendix and supplementary materials, show that the future trajectory generated by T-Diff provides effective guidance for the agent and exhibits scalability and generalizability across different simulators.

## Acknowledgements

This work was supported by the National Natural Science Foundation of China under Grant 62125207, 62032022, 62272443 and U23B2012, in part by Beijing Natural Science Foundation under Grant JQ22012 and L242020.

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

Table 6: Chosen object categories in Gibson [47] and MP3D [3].

| Dataset | Training | Evaluating |
|---------|----------|------------|
| Gibson | chair, couch, potted plant, bed, toilet, dining-table, tv, oven, sink, refrigerator, book, clock, vase, cup, bottle | chair, couch, tv, bed, toilet, potted plant |
| MP3D | char, table, picture, cabinet, cushion, sofa, bed, chest of drawers, plant, sink, toilet, stool, towel, tv monitor, shower, bathtub, counter, fireplace, gym equipment, seating, clothes | |

Table 7: FLOPs(Floating Point Operations) of SemExp [4], PONI [31], 3D-aware [60] and our T-Diff.

| Method | SemExp | PONI | 3D-Aware | T-Diff (Ours) |
|--------|--------|------|----------|----------------|
| FLOPs (M) | 3080.41 | 46621.90 | 11617.01 | 16078.40 |

# A   Appendix / supplemental material

## A.1   Experiments Setup

**Evaluation metrics.** We employ SR, SPL and DTS metrics to evaluate the ObjectNav performance and assess the trajectory generation accuracy by MSE metric.

**SR (Success Rate).** SR measures the success rate of the agent in successfully finding the target object. It is defined as $SR = \frac{1}{N}\sum_{i=1}^{N} S_i$, where $N$ is the total number of validation episodes and $S_i$ is an indicator that representing whether the $i$-th episode is successful or not.

**SPL (Success weighted by Path Length).** SPL considers whether the path length of navigation is efficient based on success rate which is formulated as $SPL = \frac{1}{N}\sum_{i=1}^{N} S_i \frac{l_i^*}{max(l_i, l_i^*)}$, where $l_i^*$ means the shortest path length calculated by the simulator and $l_i$ refers the actual path length of $i$-th episode.

**DTS (Distance to Goal).** DTS evaluates the distance $L_{i,g}$ of the agent towards the target object when the episode ends. It can be calculated as $DTS = \frac{1}{N}\sum_{i=1}^{N} max(L_{i,g} - \xi, 0)$, where success threshold $\xi = 1m$. DTS is 0 when an episode is successful.

**MSE (Mean Squared Error).** MSE assesses the accuracy of the generated trajectory which is defined as $MSE = \frac{1}{n}\sum_{i=1}^{n}(x_i - \hat{x}_i)^2$, where $n$ is the number of validation set of our collected trajectory data pairs. $x_i$ refers the ground truth trajectory and $\hat{x}_i$ is the generated trajectory.

**Object Categories.** Our experimental setup follows previous settings [31, 60, 4], and the adopted object categories are detailed in Tab. 6. For Gibson, we choose 15 categories for training and 6 for evaluating. For MP3D, there are 21 categories for both training and evaluating.

**More Visualizations.** Fig. 6 illustrates more visualizations of the generated trajectory by our T-Diff during navigation process. As shown in Fig. 6, T-Diff precisely generates the agent's future trajectory conditioned on the current state, thus effectively steering the agent towards the target object. The accuracy of the generated future trajectory reduces unnecessary exploration and guides the agent along an almost optimal path, ulimately navigating it to stop directly in front of the target object, demostrating that our T-Diff significantly improves the efficiency of navigation.

Furthermore, we provide a video demo that presents a more intuitive view of the process of trajectory denoising and navigation. Please refer to the MP4 file in the supplements zip archive.

## A.2   Computation Complexity

To compare the computational complexity of our T-Diff with the modular baselines(SemExp [4], PONI [31] and 3D-aware [60]), we utilize FLOPs(Floating Point Operations) metric to assess the computational complexity, as shown in Tab. 7, where a higher value of FLOPs refers greater computational complexity. Note that, T-Diff iterates $\tau_{max} = 100$ steps for each time step to generate the denoised future trajectory, and this trajectory is generated every $t_{T-diff} = 5$ steps during navigation. Thus, we compute the computation complexity of 100 iterations and average it over every 5 steps. The results in Tab. 7 indicate that the computation complexity of our T-Diff is higher than

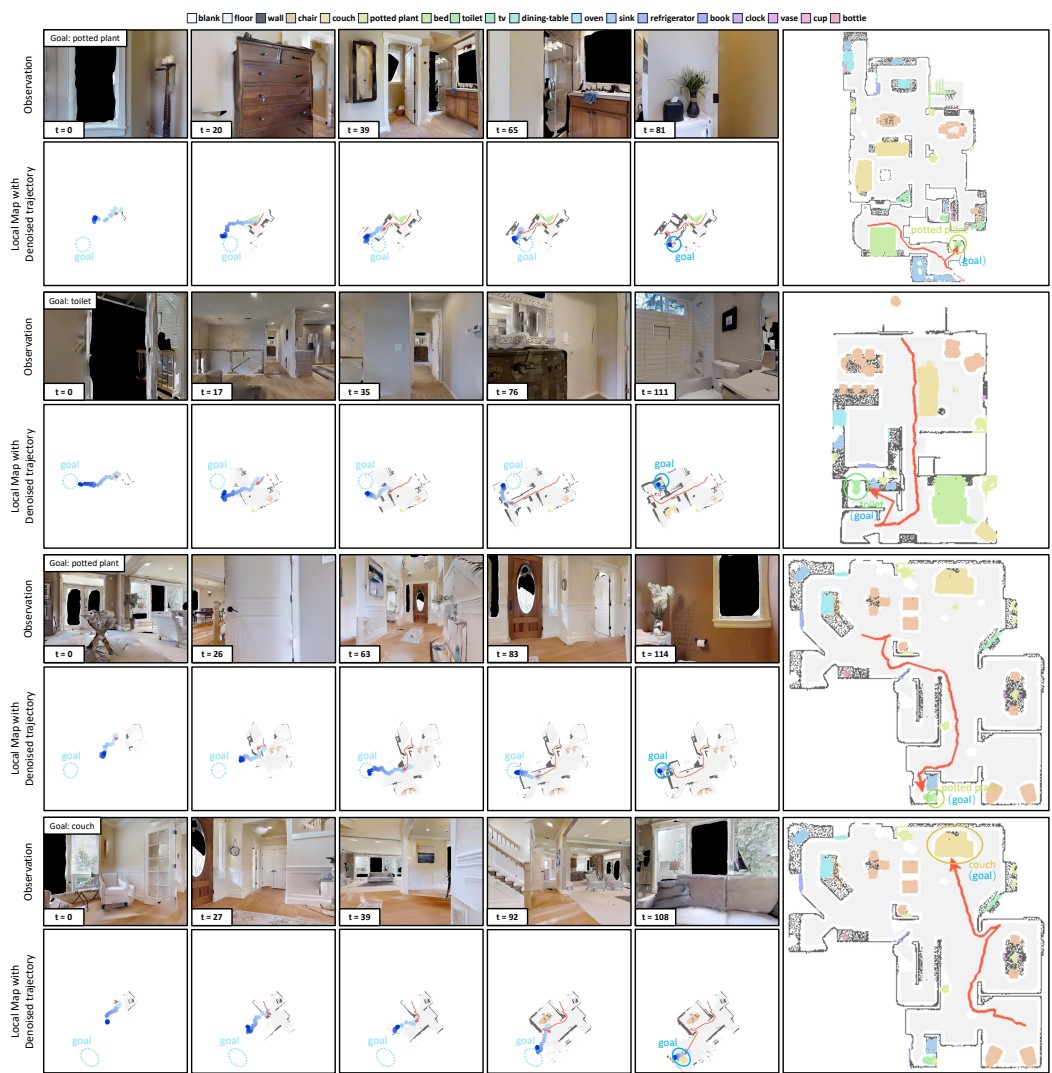

Figure 6: More visualizations. At each timestep, we visualize RGB observation, local semantic map along with the previous path and the generated future trajectory. Note that the target object location is marked with a circle.

that of SemExp, comparable to that of 3D-aware, but significantly lower than that of PONI. Although T-Diff requires multiple iteration, its operating latents is relatively small (when $k = 32$, the input dimension is $B \times 2 \times 32$) compared to PONI's input (the input dimension is $B \times (4 + n) \times 480 \times 480$, where $n$ refers to the number of semantic categories). Therefore, the compuation complexity of T-Diff is considered acceptable.

### A.3 Limitations

(1) The method of generating future trajectories based on diffusion has prediction biases. The proposed trajectory diffusion model primarily considers the current semantic map, historical trajectory information, and target category. However, other information such as the egocentric view and the relationships between object semantics, may help generate better trajectories for the ObjectNav task. Therefore, in future work, we plan to apply more conditions into the trajectory diffusion model.

(2) The training data for the trajectory diffusion model mainly comes from the collection of optimal paths, which are relatively straight and short. The types of training data pairs are not yet diverse enough, and the collection of complex trajectories requiring multiple turns and detours is still

insufficient. In future work, we plan to introduce data pairs of complex paths to enhance the model's applicability to more complex scenarios.

### A.4 Broader Impacts

Our trajectory diffusion model is a general method applied to the ObjectNav task. Although our training and testing are currently limited to the simulator stage, the trajectory diffusion model can be deployed on real robots. However, prediction errors in the model may lead to incorrect actions by the robot, potentially causing damage to personal or social property. Therefore, it must be used cautiously to ensure safety in real-world applications.

### A.5 Data License

We use two datasets (Gibson [47] and MP3D [3]), and employ Habitat simulator. All of them are published on the official papers with no licenses stated on their papers and websites. Thus, we just cite all corresponding papers without licenses.

