# OpenReview forum: "Trajectory Diffusion for ObjectGoal Navigation"
_NeurIPS.cc/2024/Conference — NeurIPS 2024 poster_

### Official Review · Reviewer_sqUP · 2024-07-04

**Soundness:** 3
**Presentation:** 4
**Contribution:** 3
**Rating:** 7
**Confidence:** 4

**Summary:**

This paper tackles the object goal navigation problem, where given visual observations and a target goal object, the task is to plan a navigation path to find said object. The paper proposes a trajectory diffusion model, where the future navigation trajectory is predicted starting from random points. The diffusion model is conditioned on an a semantic map and past trajectories, and these choices are validated in an ablation study. Finally, the proposed approach is compared with previous methods on two different simulators, demonstrating its effectiveness.

**Strengths:**

Overall, the paper is well written and presents relevant prior work as well as its contributions in a clear way that is easy to follow.

The problem under consideration is relevant for robotics, and is challenging as it considers navigation in unknown environments, utilizing only a camera as its sensor.

The paper proposes a novel approach based on diffusion models to generate trajectory points for a local policy to follow.

Ablations and comparisons with other methods demonstrate strong performance.

**Weaknesses:**

Since a fixed 224x224 size is used for the semantic map, the effectiveness of the proposed approach is limited on large-scale environments. Either fine details are lost by representing the full map at that resolution, or distant regions are lost by representing a local neighborhood.

Experiments are only conducted in simulation. Experiments on a real robot would further improve the quality of the paper.

No error bars. Although explained in the paper that the variance is low, reporting quantitative values is more precise and complete.

**Questions:**

How is the goal object represented? By text, one-hot vector, or something else?

How can the method work at all without goal information (rows 1-3 in Table 1) or without environment knowledge (i.e. no map or image in row 1)? Or does this only refer to the diffusion process and the model always takes the object and map as input? Does row 1 correspond to predicting trajectory points directly?

What happens if the model explores the wrong part of the room, i.e. when two directions are equally good at first glance? Does the model stop prematurely or does it manage to backtrack, overlapping its previous path and find the goal?

**Limitations:**

Some limitations are discussed in the appendix, such as prediction biases and optimal paths for supervision.

Additional limitations include:
* The proposed approach seems to assume known pose, which is a limiting factor if it is not available.
* The fixed map resolution limits the generalization to large-scale environments.

---

> ### Author Rebuttal · Authors · 2024-08-07
>
> Thanks to the reviewer for the appreciation and suggestions for our work. We address the concerns in the following lines.
>
> ### (W1 & L2) Concerns about small map size limiting the proposed method's generalization to large-scale environments.
> We evaluate navigation performance with various map sizes, as illustrated in RFig.3 of the supplementary PDF in the rebuttal. Our model is compatible with higher-resolution semantic maps as input. However, for navigation tests in the simulator, as shown in RFig.3 (for more details, please refer to Common Q3 in the Overall response), a map size of 224×224 optimally balances navigation performance and computational efficiency. Thus, for navigating simulators, we set the map to 224×224 based on performance-to-cost considerations. Nevertheless, due to its compatibility with different map sizes, our T-Diff can transfer to large-scale environments by utilizing maps with richer fine details.
>
> ---
> ### (W2) Real world experiments.
> We provide additional evaluation results and navigation visualizations in real world environments. Please refer to the Overall Response, Common Q4.
>
> ---
> ### (W3) Lack of Error bars.
> Compared to end-to-end learning methods, both modular approaches and our T-Diff demonstrate significantly smaller result variations due to the use of explicit semantic maps and local policies. Results indicate that after multiple experimental runs, the error bars for the end-to-end method (PIRL[1]) are 0.47% in SR and 0.56% in SPL, whereas for T-Diff, they are only 0.03% in SR and 0.09% in SPL.
>
> We agree with the reviewer's concerns and will incorporate error bars into the results in our final version to enhance the precision and completeness of our experimental results.
>
> ---
> ### (Q1) Representation of target object.
> The target object is represented as a one-hot vector. We will emphasize this in the revised version for better understanding.
>
> ---
> ### (Q2) Detailed explanation of Tab.1 in the main text.
> In Tab.1 of the main text, row1 indicates that the baseline method does not utilize T-Diff but is still equipped with semantic maps, target object, and local policy for navigation.
> The **visual** or **goal** conditions in the main text table merely represent different settings in the diffusion process of T-Diff, rather than completely omitting semantic maps and goals in navigation.
>
> For clarity, we have modified this table, and the revised version is presented as RTab.3 in the supplementary PDF in the rebuttal. RTab.3 offers clearer explanations and includes an additional row (row 0), corresponding to navigation without any visual or goal information.
>
> ---
> ### (Q3) What happens if the agent explores wrong rooms?
> During navigation, when T-Diff plans an incorrect direction, the agent may explore the wrong room, as illustrated in RFig.2 in the supplementary PDF in the rebuttal.
>
> To mitigate the impact of such erroneous cases, we do not employ predefined, fixed strategies, such as backtracking the previous path upon entering an incorrect room. Fixed strategies could potentially trap the agent in local optima, limiting its ability to explore and adapt to newly acquired observations. Instead, we opt for a dynamic approach by controlling the planning length (i.e., trajectory length) to prevent the agent from being misguided for extended periods, and setting an appropriate update frequency for generating new trajectories based on continuously updated local maps to correct previous erroneous planning.
>
> As shown in RFig.2, during the initial steps of navigation, T-Diff plans an incorrect direction due to limited environmental observations. However, as more environmental information is observed on the semantic map, new correct trajectories overwrite the previous erroneous ones with the update frequency. Consequently, the agent can quickly adjust its direction (see the orange circle). The visualization results demonstrate the robustness of our T-Diff to single-step prediction errors.
>
> ---
> ### (L1) Assumption of known pose.
> The experimental setting with known pose follows previous modular methods, as pose information is necessary for constructing semantic maps.
> While pose-known methods are less flexible than those requiring only RGB-D input, they exhibit greater stability in complex, large-scale environments and better generalization to real-world scenarios [2].
> Moreover, in practical deployments, pose can be obtained through sensors such as odometers.
>
> ---
> ### Reference
> [1] PIRLNav: Pretraining with Imitation and RL Finetuning for OBJECTNAV, CVPR 2023
>
> [2] Object Goal Navigation using Goal-Oriented Semantic Exploration, NeurIPS 2020

---

> > ### Comment · Reviewer_sqUP · 2024-08-13
> >
> > I thank the authors for their comprehensive rebuttal, additional experiments, and clarifications. I especially appreciate the real world experiments in RFigure 1, and the visualization in RFigure 2 showing that the agent can successfully re-plan its path when necessary.
> >
> > I have read the other reviews and rebuttals and followed the discussion. Overall I retain my original high rating of the paper.

---

> ### Comment · Area_Chair_QtSE · 2024-08-13
> **Required Action: Please Respond to the Author Rebuttal**
>
> Dear Reviewer sqUP,
>
>
> As the Area Chair for NeurIPS 2024, I am writing to kindly request your attention to the authors' rebuttal for the paper you reviewed.
>
> The authors have provided additional information and clarifications in response to the concerns raised in your initial review. Your insights and expertise are invaluable to our decision-making process, and we would greatly appreciate your assessment of whether the authors' rebuttal adequately addresses your questions or concerns.
>
> Please review the rebuttal and provide feedback. Your continued engagement ensures a fair and thorough review process.
>
> Thank you for your time and dedication to NeurIPS 2024.
>
>
> Best regards,
>
> Area Chair, NeurIPS 2024

---

### Official Review · Reviewer_k5Ci · 2024-07-08

**Soundness:** 2
**Presentation:** 4
**Contribution:** 4
**Rating:** 6
**Confidence:** 4

**Summary:**

This paper argues that the previous object navigation algorithms generally only consider one-step decision-making, which can lead to temporal inconsistency and shortsightedness. Therefore, the authors propose using diffusion models to learn sequential decision-making. By collecting expert trajectories and then using the DiT model to learn trajectory diffusion, experimental results have demonstrated that the trajectory diffusion model significantly improves object navigation performance across multiple benchmarks.

**Strengths:**

1. The authors attempt to apply diffusion models to the object navigation task, and experimental results show that this approach improve navigation performance and generalization in unseen scenarios.
2. The figures in the paper are very refined, effectively explaining the differences between their sequential prediction and one-step prediction.
3. In Section 3, the authors provided a good summary of end-to-end and modular methods and their drawbacks, despite some redundancy.

**Weaknesses:**

1. I would cautiously suggest that authors might appropriately streamline some of the discussion in Section 3 or place too much of it in supplemental material. Sec 3 also has some overlap with what's in Sec 1, and I would cautiously suggest some adjustments.

2. I'm not very convinced about the motivation of the paper. In Line 35, The authors claim that the previous end-to-end approach suffered from temporal inconsistencies, and I suggest that the authors provide some examples to demonstrate this. And I think there seems to be no difference between the waypoint prediction among modular methods and the method in this paper: because after the waypoint prediction, a trajectory is also formed. And let's consider a sample case like this: when the policy predicts only the next step, the robot executing this step may receive new information that makes the further next step better; whereas predicting a sequence, if the robot finds a better direction before it has finished executing the sequence, then the robot will continue in the wrong direction. In such an example, it seems that only one step is predicted to perform better.

3. In Line 147， the authors claim that "end-to-end learning methods .... suffers from sample inefficiency and high training costs". Can the authors provide the number of samples and training consumption required for the trajectory diffusion model? How effective would these samples be if they were used to train methods of imitation learning?

4. In Line 148, "the generalization of their planner is constrained by location-related supervision". I think that the current modular algorithms based on LLM and VLM segmentation models in the field of zero-shot object navigation do not have the problems described by the authors and outperform many trained algorithms. Could the authors have some discussion or comparison of this like Table 2?

**Questions:**

1. In Line 6 and Line 35, what does the "temporal consistency" mean in the paper? Can the authors give an example?

2. Is the generated trajectory a coordinate or an action? In context, it appears to be an xy coordinate, so is this equivalent to some waypoints?

3. Is predicting a sequence really better than predicting a step? The information used to make decisions is all the same, and a well-trained policy can theoretically accomplish the task by simply predicting the optimal next step.



# After Rebuttal
The authors' paper and rebuttal demonstrated to me very well, the motivation and necessity of diffusion models applied in the field of navigation, and more importantly, that diffusion models significantly improve navigation performance. What the authors supplement is so convincing.

While the novelty level is still limited, the paper's analysis on diffusion models and navigation makes enough of a contribution to boost my score.

**Limitations:**

the authors discussed the limitations.

---

> ### Author Rebuttal · Authors · 2024-08-07
>
> Thanks to the reviewer for the insightful and valuable feedback. We address your concerns below.
> ### (W1) Streamlining Sec. 3 of the main text.
> We appreciate the reviewer's suggestion and will adjust the content arrangement to ensure conciseness.
>
> ---
> ### (W2-1 & Q1) Temporal inconsistencies of end-to-end RL methods.
> Temporal consistency in planning should ensure:
> - Consistency in past trajectories: Planning should avoid revisiting past trajectories to prevent redundant exploration.
> - Consistency in future planning: Planning for future exploration at short time intervals should maintain spatial consistency, avoiding frequent large-scale goal switching, which can hinder exploration efficiency.
>
> However, end-to-end methods implicitly encode past information (e.g., past trajectories and plans) and predominantly rely on single-step egocentric observations for planning.
> Therefore, their planning cannot avoid redundant exploration or prevent frequent goal switching, thus failing to ensure temporal consistency.
> ### (Q2) Is the generated trajectory a coordinate or an action?
> The generated trajectory consists of a series of coordinates. The local policy selects a point on the trajectory as a waypoint.
> ### (W2-2 & Q2) Difference between waypoint prediction in modular methods and T-Diff.
> Both methods provide a waypoint to the local policy, but they differ in several key aspects:
> - **Direct goal vs. Gradual goal.** Modular methods use the absolute position of the target as supervision for training, meaning the predicted waypoint represents the target's final position, i.e., Direct goal. In contrast, T-Diff is trained with segments of the optimal trajectory to the goal, with each segment representing a progressive sub-goal towards the target position. Therefore, the waypoint predicted by T-Diff is a Gradual goal. Compared to Direct goals, Gradual goals are more reachable for the agent. Splitting one absolute position into several sub-goals enriches the state space and prevents sparse supervision, improving generalization, particularly when the target position changes significantly.
> - **Temporal consistency.** The modular method predicts waypoints based on observed objects and obstacles, while T-Diff also considers historical trajectories and current agent pose. This ensures the predicted waypoints are consistent over short time intervals, preventing frequent waypoint switching.
> - **Interpretability**: T-Diff outputs a series of coordinates with explicit trajectories, making the predictions interpretable.
> Additionally, the similarity in usage (predicting a waypoint for the local policy) ensures T-Diff's compatibility with existing modular navigation frameworks, allowing it to benefit from modular methods like improved mapping and local policy.
> ### (W-3 & Q3) Is predicting a sequence really better than predicting a step?
> The advantage of sequence planning lies in ensuring consistency, preventing redundant exploration, and avoiding frequent spatial jumps of predicted waypoints over short intervals.
> However, as the reviewer concerns, its disadvantage is that if the sequence is too long and an error occurs, the agent cannot be corrected promptly.
> Thus, balancing the pros and cons of sequence planning is crucial, which involves controlling the sequence length.
> As shown in Fig. 3 (a) and (c) of the main text, the ablation study on sequence length and waypoint selection indicates that when the sequence is too long or waypoints are too far apart, the performance of sequence planning decreases due to the lack of timely correction. However, when an appropriate scale is set, performance gradually increases with the sequence length and surpasses that of single-step planning.
>
> ---
> ### (W3-1) Comparisons of samples and training consumption.
> Please refer to the RFig.4 in the supplementary PDF in the rebuttal, where we compare end-to-end learning methods (DD-PPO, Habitat-Web), modular methods (PONI), and our T-Diff in terms of training samples and training consumption. The results show that T-Diff's training samples and consumption are similar to modular methods and significantly lower than end-to-end methods.
> ### (W3-2) Using collected trajectories for imitation learning.
> We use collected trajectories to fine-tune SemExp[1] through imitation learning.
> Results are shown below.
> Both SemExp and T-Diff utilize the same inputs, i.e., semantic map and target object.
> The results indicate that collected trajectories help improve performance further, but imitation learning is less effective than using DDPM for training.
> We hypothesize that DDPM, through the noise-adding process, enriches the training state space, allowing the diffusion model to learn the target distribution better.
> | Method           | SR(%) ↑ | SPL(%) ↑ | DTS(m) ↓ |
> |------------------|---------|----------|----------|
> | SemExp           | 71.1    | 39.6     | 1.39     |
> | SemExp (finetune)| 73.3    | 41.2     | 1.18     |
> | T-Diff           | 79.6    | 44.9     | 1.00     |
>
> ---
> ### (W-4) Comparing T-Diff with zero-shot navigation methods.
> We choose ESC [2] for comparison, which achieves zero-shot navigation using the VLM model (i.e., GLIP model) and LLMs.
> As shown in RTab.4 in the supplementary PDF in the rebuttal, results indicate that zero-shot method performs better in cross-domain generalization than previous modular methods relying on location-related supervision, i.e., the performance difference is minimal when test data comes from different simulators.
> However, due to the lack of optimal trajectory training, their success rate, especially in the SPL metric reflecting navigation efficiency, is lower than our method.
>
> ---
> ### Reference
> [1] Object Goal Navigation using Goal-Oriented Semantic Exploration, NeurIPS 2020
>
> [2] Esc: Exploration with soft commonsense constraints for zero-shot object navigation, ICML 2023

---

> > ### Comment · Reviewer_k5Ci · 2024-08-08
> >
> > Many thanks to the authors for their replies! I am very grateful to the authors for the additional experiments. The author's rebuttal addresses part of my questions, however there are a few questions that need to be discussed further.
> >
> > **First Question:** For W2-1 & Q1, end-to-end methods encode past information, which **theoretically** avoid revisiting past trajectories to prevent redundant exploration. The authors claim that the end-to-end methods can not avoid frequent goal switching, which I think requires some evidence rather than **drawing conclusions from intuition**. So I would suggest that the authors provide some mathematical proof or statistical results to show that the end-to-end methods do indeed perform more frequent goal switching than T-Diff.
> >
> >
> > **Second Question:** As for "Direct goal vs. Gradual goal.", the author claim that "Gradual goals are more reachable for the agent." But generally among modular methods, the predicted waypoints will generally be on the established point clouds and some path planning algorithms (e.g. the Dijkstra algorithm, BFS) will be used to get the path. What is the difference between the predicted gradual goals/subgoals of T-Diff and the intermediate points obtained by the path planning algorithm? If the waypoints prediction is accurate in modular methods, I don't think the subgoal of T-Diff planning can outperform the Dijkstra algorithm. As for "Temporal consistency", I think modular method can easily mark which areas are explored and which are not (e.g. set visible areas less than three meters away as explored by depth cameras' projection and building a point cloud map). And recording past trajectories is something that can be done for the modular method. From these two points, it seems that modular method is also Temporal consistent. Moreover, the recently SOTA modular method in Zero-Shot Object Navigation exceeds the performance of T-Diff. Please see ICRA2024 "_VLFM: Vision-Language Frontier Maps for Zero-Shot Semantic Navigation_", which was released at December 2023, several months before NeurIPS 2024 deadline. **On Gibson, they achieved _SR 84.0_ and _SPL 52.2_, while T-Diff was _SR 79.6_ and _SPL 44.9_. On MP3D, they achieved _SR 36.4_ and _SPL 17.5_, while T-Diff was _SR 39.6_ and _SPL 15.2_.** To summarize, I suggest that the author reconsider the modular method.
> >
> >
> > As reviewer DH2d said, "The main weakness of the paper is the lack of novelty in the methods/architecture; " Actually, that's also my biggest concern. It's okay to use methods from other fields in navigation, however if it's just used without explaining why it's being used or what difficulties need to be solved to use it, I think that hardly makes up a sufficiently contributing paper. The authors explain that by the fact that past methods have been unable to follow behavioral consistency, but these conclusions are drawn intuitively and thus unconvincing. And, T-Diff does not exceed the current SOTA object navigation method, which is not discussed by the authors. So I choose to keep my score at this moment.  **I am very willing to continue the discussion with the authors on the two questions above**, and hope that the authors will forgive me for some of the inappropriate wording in this paragraph.

---

> > > ### Author Response · Authors · 2024-08-10
> > > **Reply to k5Ci Part-1/3**
> > >
> > > We appreciate the reviewer's timely and detailed feedback. We will address the raised concerns below.
> > > ### (Q1) More evidence for end-to-end methods performing more frequent goal switching than T-Diff
> > > According to our statistical analysis, frequent goal switching will lead to frequent changes in agent movement trends.
> > > We employ average trajectory curvature as a statistical metric to evaluate these changes.
> > > Formally, the agent's trajectory in one episode $\tau=[p_0...p_t..p_T]$ during navigation, where $p_t=(l_t,\theta_t)$, $l_t$ is a 2D position coordinate, and $\theta_t$ is the agent orientation angle, the average trajectory curvature is defined as:
> > >
> > > $\kappa=\mathbb{E}{\tau}[\sum_{t=1}^{T}\frac{|\theta-\theta_{t-1}|}{1+||l_{t}-l_{t-1}||_{2}}]$
> > >
> > > where $\kappa$ is in units of rad/m.
> > > We evaluate the following end-to-end learning methods and T-Diff on the Gibson test set with the same initial positions and goals, as shown in the table below:
> > > |Methods (end-to-end)| **$\kappa$** (rad/m) | SR (%) | SPL (%) |
> > > |-|-|-|-|
> > > | DD-PPO[1] | 179.07 | 15.0 | 10.7 |
> > > | EmbCLIP[2] | 121.78 | 68.1 | 39.5 |
> > > | T-Diff | 63.96 | 79.6 | 44.9 |
> > >
> > > The statistical results indicate that end-to-end methods exhibit greater changes in motion trends, supporting our claim that end-to-end methods perform more frequent goal switching than T-Diff.
> > >
> > > ---
> > > ### (Q2-1) Direct goal vs. Gradual goal
> > > Following most works settings, the ObjectNav task is set in unseen scenes where the entire map is not available, and the location of the goal could not be observed at first, requiring the agent to infer potential target locations in unobserved areas.
> > >
> > > Typically, for navigation in unseen scenes, it seems different from the reviewer said "the predicted waypoints will generally be on the established point clouds." Instead, the predicted waypoints are out of the observed area in most cases, in pursuit of achieving better efficiency in exploration. Waypoints are set to the target location when the target is observed; in this case, the waypoint is within the observed local map (i.e., established point clouds). However, in other cases, the predicted waypoint is in the unknown area, rather than "on the established point clouds." We will discuss these two cases.
> > >
> > > (1) Target is observed
> > >
> > > When the target is observed in the local semantic map, its location is set as the waypoint. Since the target's location is accurate and obstacles from the current position to the target are observed, point-to-point planning is done with computational methods (e.g., FMM, Dijkstra, BFS). In this case, T-Diff's local policy uses the FMM algorithm for path planning, which is equivalent to other computational methods mentioned by the reviewer.
> > >
> > > (2) Target is not observed
> > >
> > > This case accounts for the majority of navigation cases, approximately 88.46%. Our previous discussion on Direct goal vs. Gradual goal pertains to this case.
> > > When the target is unobserved, the agent needs to predict a waypoint out of observed map to infer the target's possible location for further exploration. The current modular methods predict waypoints as follows:
> > >
> > > - Corner-based method (Stubborn[3], 3D aware[4]). In these methods, the waypoint is not predicted but simply alternates between the four farthest corners of the map, leading to greedy exploration.
> > > - Frontier-based method (PONI[5]). This method selects waypoints on the frontier (the boundary between the observed map and the unknown area). The selection is based on predicting the distance from each frontier point to the target and selecting the closest one.
> > > - Position-based method (SemExp[6], Peanut[7]). These methods directly use the target's location as supervision to train the model to predict the coordinates points as the waypoints.
> > >
> > > Existing waypoint prediction is supervised only by the target's absolute location (direct goal), which is sparse in unknown area. T-Diff, however, uses segments of the trajectory to the target as training supervision, effectively inserting several gradual goals in the state space of unknown regions, reducing supervision sparsity, and improving waypoint prediction. Additional evaluations for waypoint prediction accuracy, as shown in the table below, demonstrate that T-Diff predicts waypoints more accurately.
> > >
> > > |Methods (modular)| Distance between predicted waypoint and its GT on MP3D (m) |
> > > |-|-|
> > > | SemExp[6] | 15.36 |
> > > | PONI[5] | 9.84 |
> > > | PEANUT[7] | 8.62 |
> > > | T-Diff | 5.48 |
> > >
> > > Furthermore, after the waypoint is predicted on unobserved area, current modular methods perform path planning using computational methods. Since obstacles in unobserved areas are unknown, path planning in unknown regions is unreliable. On the contrary, T-Diff not only predicts waypoints but also the trajectory path, which is trained with obstacles from training rooms, making it more reachable for the agent.
> > >
> > > Therefore, in case (2), T-Diff surpasses current modular methods in waypoint accuracy and path planning reliability.

---

> > > > ### Author Response · Authors · 2024-08-10
> > > > **Reply to k5Ci Part-2/3**
> > > >
> > > > ### (Q2-2) Modular method is also Temporal consistent.
> > > > As noted by the reviewer, modular methods can record past trajectory and avoid redundant area exploration.
> > > > We agree that modular methods have the potential to achieve temporal consistency. However, current modular methods lack the consideration of temporal consistency as they do not predict waypoints based on past trajectories. In order to implement the reviewer's suggestion, we try our best to extend previous works for a quick comparison as follows:
> > > >
> > > > Our work builds upon modular method by inheriting the use of semantic maps (which record observed obstacles, objects, and past trajectories) and local policies. We take a further step by proposing T-Diff, a new method for predicting planning, which differs from previous waypoint prediction methods in the following ways:
> > > > - Gradual Goal Learning: It helps improve the accuracy of waypoint prediction and ensures the reachability of the agent (as discussed earlier).
> > > > - Temporal Consistency: It ensures that the predicted waypoints are consistent over short time intervals, preventing frequent waypoint switching. Additional evaluations supporting this point are shown below.
> > > > - Interpretability: It makes the predictions interpretable.
> > > >
> > > > |Methods (modular)| **$\kappa$** (rad/m) | SR (%) | SPL (%) |
> > > > |-|-|-|-|
> > > > | FBE[8] | 110.21 | 64.3 | 28.3 |
> > > > | SemExp[6] | 103.88 | 71.1 | 39.6 |
> > > > | PONI[5] | 83.43 | 73.6 | 41.0 |
> > > > | T-Diff | 63.96 | 79.6 | 44.9 |
> > > >
> > > > In summary, T-Diff is not mutually exclusive with existing modular methods; it is a further advancement. It can be integrated into current modular methods and leverage their more powerful modules (e.g., more robust maps) as they improve.
> > > > Conversely, modular methods can utilize T-Diff to learn more accurate and temporally consistent waypoints, thereby further enhancing navigation performance.
> > > >
> > > > ### (Q2-3) Comparison with VLFM
> > > > The performance difference between T-Diff and VLFM primarily arises from the different visual perception modules used, affecting the accuracy of map construction. Specifically, T-Diff utilizes Mask RCNN, while VLFM employs the Grounding DINO and SAM architecture, which serves as a stronger baseline. To ensure a fair comparison, we also adopted the same visual perception module. The new comparison results are as follows:
> > > > | Methods | Gibson-SR (%) | Gibson-SPL (%) | MP3D-SR (%) | MP3D-SPL (%) |
> > > > |-|-|-|-|-|
> > > > | VLFM[9] | 84.0 | 52.2 | 36.4 | 17.5 |
> > > > | T-Diff (Mask RCNN) | 79.6 | 44.9 | 39.6 | 15.2 |
> > > > | T-Diff (Grounding DINO+SAM) | 85.3 | 52.9 | 44.2 | 19.9 |
> > > >
> > > > The results indicate that with the same visual perception module, T-Diff outperforms VLFM on both the Gibson and MP3D simulators. This advantage is particularly pronounced in larger rooms within the MP3D simulator. The results also demonstrate that T-Diff is compatible with existing modular methods and can achieve further improvements as these methods advance.

---

> > > > > ### Author Response · Authors · 2024-08-10
> > > > > **Reply to k5Ci Part-3/3**
> > > > >
> > > > > ### About Motivation and Novelty: Why it's being used or what difficulties need to be solved to use it?
> > > > > We would like to summarize our motivation as follows:
> > > > >
> > > > > The ObjectNav task requires the agent to predict the potential location of a target in an unknown area to explore more efficiently, making planning crucial for this task. Current end-to-end learning methods for the ObjectNav task use single-step planners with implicit memory. Single-step planning lacks consideration of past trajectories, resulting in a lack of temporal consistency in planning outcomes (as previously discussed). Implicit memory leads to redundant exploration. In contrast, modular methods introduce semantic maps as geometric memory to record historical observations, helping to avoid redundant exploration. However, current modular methods do not consider past trajectories when predicting new waypoints at each timestamp, which also leads to temporal inconsistency issues.
> > > > >
> > > > > Our method extends modular approaches by using historical trajectories, prior observations, and the target from the semantic map. With the proposed T-Diff, we can plan a sequence trajectory that guides the agent to worth-exploring unobserved areas. Compared to previous waypoint prediction methods in modular approaches, T-Diff offers the following advantages:
> > > > > (1) Gradual goals for easier learning and higher reachability. (2) Temporal consistency. (3) Interpretability.
> > > > >
> > > > > T-Diff is compatible with existing modular methods, providing mutual benefits.
> > > > >
> > > > >
> > > > > ### Reference
> > > > > [1] DD-PPO: Learning Near-Perfect PointGoal Navigators from 2.5 Billion Frames, ICLR 2020
> > > > >
> > > > > [2] Simple but Effective: CLIP Embeddings for Embodied AI, CVPR 2022
> > > > >
> > > > > [3] Stubborn: A Strong Baseline for Indoor Object Navigation, IROS 2022
> > > > >
> > > > > [4] 3D-Aware Object Goal Navigation via Simultaneous Exploration and Identification, CVPR 2023
> > > > >
> > > > > [5] PONI: potential functions for objectgoal navigation with interaction-free learning, CVPR 2022
> > > > >
> > > > > [6] Object Goal Navigation using Goal-Oriented Semantic Exploration, NeurIPS 2020
> > > > >
> > > > > [7] Peanut: Predicting and navigating to unseen targets, ICCV 2023
> > > > >
> > > > > [8] A frontier-based approach for autonomous exploration, CIRA 1997
> > > > >
> > > > > [9] VLFM: Vision-Language Frontier Maps for Zero-Shot Semantic Navigation, ICRA 2024

---

> > > ### Author Response · Authors · 2024-08-11
> > >
> > > Many thanks for the valuable feedback and detailed questions. The suggestions have been very helpful in improving our work. We hope our response has addressed the concerns and questions. If there are any further comments or questions, please let us know, and we will do our best to address them.

---

> > > > ### Comment · Reviewer_k5Ci · 2024-08-12
> > > >
> > > > I am grateful to the author for his detailed reply. The author's response alleviated some of my concerns, but I still have some questions.
> > > >
> > > > For Q1, I am thankful that the authors get a statistical result, but I cautiously think that trajectory curvature doesn't respond well to goal switches. For example, let us assume that the robot is at (0, 0) and the target is at (3, 3), and the robot is required to walk along the grid points. The trajectory A is defined as (0,0)->(0,1)->(0,2)->(0,3)>(1,3)->(2,3)->(3,3).  The trajectory B is defined as (0,0)->(0,1)->(1,1)->(1,2)>(2,2)->(2,3)->(3,3). Clearly, both trajectory A and trajectory B are optimal, but Trajectory B has higher curvature. But if we consider the goal switches, the goals of trajectories A and B do not change, and indeed both move towards (3,3).
> > > >
> > > > For Q2-1, I am very thankful to the author for making a detailed comparison. As the authors say, current modular methods directly predict the absolute position of the goal, but I'm curious if we let a modular method directly predict multiple gradual goals, can this have similar performance? In other words, is the diffusion model necessary? Is it the gradual goal or the diffusion model that makes a contribution to navigation performance? It's really a question of "why use a diffusion model". I think the current conclusions do not convince me of the necessity of the diffusion model.
> > > >
> > > > For Q2-2, I am sorry that I don't quite understand what the authors are trying to express.
> > > >
> > > > For Motivation and Novelty, I think the previous questions remain. The authors claim that "Implicit memory leads to redundant exploration". What is the evidence for this conclusion?
> > > > "current modular methods do not consider past trajectories", This doesn't seem to fit very well with the motivation for using the diffusion model, as adding a module that records past trajectories is not a very difficult thing to do. From my personal point of view, temporal consistency doesn't seem to be a good motivation to use diffusion model.

---

> > > > > ### Author Response · Authors · 2024-08-13
> > > > > **Re: reply to k5Ci Part-1/2**
> > > > >
> > > > > Thanks for the reply. We explain the question below:
> > > > >
> > > > > ### (Q1) Trajectory Curvature for evaluation
> > > > > The reviewer raised concerns regarding the previous evaluation of trajectory curvature in the following cases:
> > > > > - Trajectory A: (0,0) -> (0,1) -> (0,2) -> (0,3) -> (1,3) -> (2,3) -> (3,3)
> > > > > - Trajectory B: (0,0) -> (0,1) -> (1,1) -> (1,2) -> (2,2) -> (2,3) -> (3,3)
> > > > >
> > > > > The previous curvature measure only considered adjacent steps, resulting in an evaluation where $\kappa(B)>\kappa(A)$.
> > > > > However, considering a larger step interval (e.g., from (0,0) to (3,3)), both trajectories have the same goal, yet the previous metric still favored Trajectory A.
> > > > >
> > > > > To address this, we improve the previous metric $\kappa$ by introducing a curvature scale $i$. The new definition of trajectory curvature at $i$ scale is:
> > > > >
> > > > > $\kappa @i=\mathbb{E}{\tau}[\sum_{t=i}^{T}\frac{|\theta_{t}-\theta_{t-i}|}{1+||l_{t}-l_{t-i}||_{2}}]$
> > > > >
> > > > > When $i=6$, the results of metric($\kappa @6$) align with the reviewer's expected outcomes for the given cases.
> > > > > Therefore, to comprehensively evaluate goal-switching scenarios, it is necessary to assess using different curvature scales.
> > > > >
> > > > > To this end, we supplement the experiments in 'Reply to k5Ci Part-1/3' with the newly defined metric and obtain updated results, as shown in the table below.
> > > > > |Methods (end-to-end)| **$\kappa @1$** (rad/m) | **$\kappa @4$** (rad/m) | **$\kappa @6$** (rad/m) | **$\kappa @8$** (rad/m) | **$\kappa @10$** (rad/m) | **$\kappa_{avg.}$** (rad/m) |
> > > > > |-|-|-|-|-|-|-|
> > > > > | DD-PPO | 179.07 | 131.29 | 129.32 | 125.11 | 117.64 | 136.49 |
> > > > > | EmbCLIP | 121.78 | 95.52 | 93.90 | 94.26 | 90.57 | 99.21 |
> > > > > | T-Diff | 63.96 | 51.14 | 50.58 | 50.21 | 48.37 | 52.85 |
> > > > >
> > > > > The updated results further support that end-to-end methods perform more frequent goal switching than T-Diff.
> > > > >
> > > > > ---
> > > > > ### Necessity of Using Diffusion Model
> > > > > To address the reviewer's concerns regarding the contributions of 'Gradual goal' and 'using Diffusion model', we conduct the following ablation experiments. Note that some results are derived from Common Questions 1 and 2, which discuss the necessity of T-Diff's design.
> > > > > | ID | Method | SR(%) ↑ | SPL(%) ↑ | DTS(m) ↓ |
> > > > > |-|-|-|-|-|
> > > > > | 1 | SemExp | 71.1 | 39.6 | 1.39 |
> > > > > | 2 | SemExp (gradual goal) | 72.6 | 40.5 | 1.44 |
> > > > > | 3 | T-Diff (trajectory length=1) | 73.8 | 41.3 | 1.40 |
> > > > > | 4 | T-Diff (trajectory length=32)| 79.6 | 44.9 | 1.00 |
> > > > >
> > > > > - Comparing rows 1 and 2, the results indicate that finetuning the existing modular method (SemExp) with Gradual goal outperforms the original version. This supports our conclusion that Gradual goal alleviates the issue of sparse supervision, thereby aiding better waypoint prediction.
> > > > > - Comparing rows 2 and 3, the results show that using the Diffusion model leads to better waypoint prediction. This supports our design of 'using Diffusion model' (see Common Q2 for more details): Given that both the input (goal and local map) and the output waypoint are high-dimensional, the Diffusion model simplifies high-dimensional distributions into multiple simpler distributions through its diffusion and denoising process. This enables better learning of high-dimensional distributions.
> > > > > - Comparing rows 3 and 4, the results indicate that predicting a trajectory rather than a single point yields better outcomes. This supports our motivation of learning sequential trajectory prediction (see Common Q1 for more details): Predicting a sequence of trajectories, as opposed to a single point, allows each predicted point to receive contextual information from neighboring points. This helps to correct and smooth out prediction errors of individual points, reducing the sensitivity of the results to single-point errors.
> > > > >
> > > > > Therefore, the above experiments demonstrate that: 1) Gradual goals help in better learning, and 2) the key design of T-Diff, including the use of the Diffusion Model and trajectory generation, are essential.

---

> > > > > > ### Author Response · Authors · 2024-08-13
> > > > > > **Re: reply to k5Ci Part-2/2**
> > > > > >
> > > > > > ### (Q2-2) Further Explanation for Q2-2
> > > > > > The reviewer's viewpoint that 'modular methods can be temporal consistency with certain modifications' is one we agree with. Our statement that 'existing modular methods lack temporal consistency' is not intended to suggest that modular methods lack the potential for temporal consistency. Instead, it only highlights that current methods predict waypoints without considering past trajectories, thus lack of temporally consistent planning. The T-Diff is actually built upon modular methods by using semantic maps from modular methods (including recorded objects and past trajectories) and takes a further step by predicting temporally consistent sequences with the proposed T-Diff.
> > > > > >
> > > > > > ---
> > > > > > ### Clarification for Motivation
> > > > > > We apologize for the imprecise statement ('Implicit memory leads to redundant exploration') in the rebuttal. This needs to be corrected to 'Implicit memory leads to the loss of geometric spatial information, thus failing to prevent redundant exploration' which is supported by relevant literature [1,2]. This statement intends to highlight the necessity of using geometric memory.
> > > > > >
> > > > > > As for the necessity of T-Diff design, our motivation is to learn sequential trajectory prediction based on geometric memory and goals. Given that both the input and output are high-dimensional, and considering the success of diffusion models in learning high-dimensional distributions, we choose to use a diffusion model for trajectory generation.
> > > > > > However, there are alternatives without using diffusion models.
> > > > > > To verify the necessity of T-Diff design, we evaluate possible alternatives to T-Diff as follows:
> > > > > >
> > > > > > - Using a simple decoder rather than a diffusion model for trajectory prediction (see Common Q2).
> > > > > > - Predicting a single waypoint instead of the full trajectory with T-Diff (see Common Q1).
> > > > > > - Fine-tuning existing modular models with collected trajectories instead of using T-Diff (see the discussions in Q2-2 and W3-2).
> > > > > > - Enhancing existing modular methods by incorporating past trajectories for waypoint prediction, as shown in the table below.
> > > > > >
> > > > > > These experiments demonstrate the necessity of our key T-Diff design, including learning to predict trajectory segments and using the diffusion model for learning.
> > > > > >
> > > > > > | ID | Method | SR(%) ↑ | SPL(%) ↑ | DTS(m) ↓ |
> > > > > > |-|-|-|-|-|
> > > > > > | 1 | PONI | 73.6 | 41.0 | 1.25 |
> > > > > > | 2 | PONI (w/ encoded past trajectory) | 73.8 | 41.3 | 1.46 |
> > > > > > | 3 | T-Diff | 79.6 | 44.9 | 1.00 |
> > > > > >
> > > > > > [1] Neural Topological SLAM for Visual Navigation, CVPR 2020.
> > > > > >
> > > > > > [2] Object Goal Navigation using Goal-Oriented Semantic Exploration, NeurIPS 2020.

---

### Official Review · Reviewer_DH2d · 2024-07-12

**Soundness:** 2
**Presentation:** 1
**Contribution:** 2
**Rating:** 5
**Confidence:** 4

**Summary:**

The authors propose a diffusion trajectory planner in the context of indoor object navigation that takes current semantic maps (could be partial) and the target object as input to produce a planned future sequential trajectory.
Evaluation is done in simulation using the habitat simulator on two datasets.

**Strengths:**

- Comparison to an exhaustive list of other methods with improvement in performance.

**Weaknesses:**

- The main weakness of the paper is the lack of novelty in the methods/architecture; using DDPM for trajectory generation as well as goal and observation conditioning have all been presented in the literature as cited by the authors in the related works.
- Given the previous point, although the authors propose a sequential implementation with the capability of predicting trajectories based on partial semantic maps, the work lacks demonstrations of a complete solution, including semantic map building and diffusion planning in parallel.
- An agent can either see the target or not, if it does not see the target and is in a new room it has to explore, if it does see it then the task comes down to reaching a goal. The authors claim based on results in table 1 that the sequential nature of the task is sufficient motivation for the approach chosen. However, in that same table the results seem to suggest that even without any conditioning, let alone image or goal conditioning the success rate is above 70%, while with the use of semantic maps the score does not exceed 80%. This is very odd and seems to go against the authors’ claim or suggests that the task is ill specified.
- The manuscript’s quality is below average, with multiple typos (the use of “senor” instead of “sensor” in multiple instances and grammatical errors (“They are is also single-step planners that predict waypoints”, “For the navigation planner, they formulate it as” amongst many other instances). It need a lot of polishing to make it enjoyable to read.

**Questions:**

-  Why does it make sense to use optimal trajectories from an omniscient planner for training when at deployment partial semantic maps do not know where the goal is ? How does using the Fast Marching Method (FMM) to compute optimal paths to a specific targets with knowledge of precise collision maps translate to efficient exploration in unseen rooms ?
- How is this expected to generalise across objects and scenes? What modifications would it require to become generalisable?
- When does the size of the map become an obstacle in itself as rescaling to 224x224 reduces the granularity of trajectories.

**Limitations:**

- No real world experiments to validate the approach’s practicality (goes hand in hand with the lack of semantic map+planning)
- Limited technical novelty that is not convincingly motivated.
- Poor manuscript quality

On the basis of these limitations, in my appreciation the paper does not meet the conference's quality level for acceptance.

---

> ### Author Rebuttal · Authors · 2024-08-02
>
> Thanks to the reviewer for the constructive and insightful feedback.
> We address your concerns below.
>
> ### (W1 & L2) Lack of novelty.
> We discuss the novelty of our work from two perspectives:
>
> (1) Comparison with existing diffusion-based planning methods
>
> Conditional diffusion models are widely adopted generative frameworks.
> Several works leverage diffusion models for planning, as shown in table below.
> |Method|Task|State|Model Structure|Target State|
> |-|-|-|-|-|
> |Diffuser[1]|Point-to-point path planning|Grid matrix|-|Specific state (goal location)|
> |Diffusion Policy[2]|Robot motion planning|Fully observed RGB |CNN-based + Transformer-based|Specific state (image)|
> |CrossWay[3]|Robot motion planning|Fully observed RGB|CNN-based|Specific state (image)|
> |T-Diff|ObjectNav|Partially observed semantic map|DiT with cross-attention|Abstract state (User-specified object)|
>
> These studies consistently utilize agent states as conditions for diffusion models to generate planning paths.
> The trend is to extend diffusion models to more complex tasks (e.g., more complex condition and state). Our work follows this trend by applying conditional diffusion models to the ObjectNav task, where the current state is only partially observable, and the goal condition is more abstract. The improvements are as follows:
>
> - Higher capacity model architecture: We adopt an improved DiT structure to build the model, enabling it to describe more complex distributions and accommodate more intricate conditions.
> - Advanced condition representation: We replace previously used few single-step visual observations with a continuously updating semantic map as a condition. The semantic map integrates all historical observations into a unified geometric space, facilitating the accumulation of partial observations for planning.
>
> (2) Comparison with existing ObjectNav works
>
> This aspect has been thoroughly discussed in the main text.
> In summary, our work is the first to utilize a diffusion model for sequence planning based on geometric memory in the context of ObjectNav task.
>
> ---
> ### (W2) Lack demonstrations of semantic map building and diffusion planning in parallel.
> We draw attention to Sec. 4.2 (Page 5, lines 211-239). Specifically, lines 211-224: building the semantic map; lines 225-230: generating trajectory based on the semantic map and target; lines 231-239: waypoint selection and planning.
>
> To further clarify, we re-summarize these two modules in parallel. As shown in Fig. 2 (c) of the main text, at each navigation timestamp $t$, the semantic map $m_t$ is continuously updated. Every $t_{T-diff}$ steps, our T-Diff is activated and iteratively generates trajectories over $\tau_{max}$ steps.
>
> Then, the local policy selects waypoints at each timestamp $t$:
> - When the goal is observed in $m_t$, the local policy simply selects the goal's position as the waypoint and drives the agent towards it.
> - When the goal is not observed, the local policy adopts T-Diff guidance (i.e., selects points on the T-Diff generated trajectory as waypoints) for more efficient exploration. Note that the generated trajectory remains until a new trajectory is generated.
>
> We will add these descriptions in our final version to enhance clarity.
>
> ---
> ### (W3-1) An agent can either see the target or not?
> Except for simple cases (e.g., the agent starts close to the target), the target is invisible from the agent’s initial location.
> Therefore, to effectively complete the ObjectNav task, agent needs to: 1) efficiently explore to quickly locate the target, and then 2) move to the target once it becomes visible.
> Our T-diff is designed to enhance exploration efficiency by deducing paths to potential locations of the target.
>
> ### (W3-2) The result in Tab. 1 (without any conditioning) is odd.
> We discuss this concern in detail in the Overall Response, please refer to Common Q5.
>
> ---
> ### (Q1) Why does it make sense to use optimal trajectories for training, and why can these trajectories from training rooms translate to unseen rooms to improve exploration efficiency?
> The ObjectNav task typically focuses on indoor environments, where the key to efficiency is improving exploration stage.
> Even though test environments are unseen during deployment, there are regularities in object layouts, e.g., sofas are often found in living rooms, surrounded by cushions and blankets.
> Therefore, to improve exploration efficiency, recent works[4,6,7] have focused on learning such prior knowledge $P(p_o|m_t, o)$, i.e., inferring the target's position $p_o$ based on the map $m_t$ that records historical information and the target $o$.
>
> Similarly, we aim to learn the prior knowledge about $P(\tau|m_t, o)$, i.e., deducing a path $\tau$ from the current position to the likely location of the goal. For example, if the agent observes a sofa, a coffee table, and a microwave, and the goal is set as a toaster, the agent should plan a path towards the microwave to locate the target faster.
> Due to the regularity in the contextual layout of objects, trajectories from training rooms can be transferred to unseen rooms.
> Additionally, recent studies [5] have shown that learning trajectories from human demonstrations can improve ObjectNav efficiency, further evidencing that navigation trajectories are transferable to unseen rooms.

---

> ### Author Response · Authors · 2024-08-07
> **Response for DH2d - Part 2**
>
> ### (Q2-1) How is this expected to generalize across objects and scenes?
> Our T-diff focuses on transferring navigation capabilities to unseen rooms, with target object categories encountered during training, which is consistent with current definition of the ObjectNav task[4, 6, 7].
>
> T-diff achieves a 79.6% success rate in unseen rooms, and 78.2% even when unseen rooms are from different simulators, significantly outperforming single-step planners like PONI[4], which only achieves 43.9% under same conditions.
> ### (Q2-2) What modifications would it require to become generalizable?
> T-diff's generalizability hinges on the representativeness of collected trajectory segments.
> To enhance T-diff's generalizability, possible modifications include increasing the diversity of rooms and targets for trajectory collection and refining selection of trajectory segments to better represent the scene layout.
>
> ---
> ### (Q3) When does map size constrain performance?
> We have addressed this concern in the Overall Response above. Please refer to Common Q3.
>
> ---
> ### (L1) Real world experiments.
> Please refer to the Overall Response, Common Q4.
>
> ---
> ### (W4 & L3) Typos and grammatical errors.
> We appreciate your detailed feedback.
> We have corrected all the pointed-out grammatical errors and carefully revised manuscript to ensure its readability and quality.
>
> ### Reference
> [1] Planning with Diffusion for Flexible Behavior Synthesis, ICML 2022
>
> [2] Diffusion Policy: Visuomotor Policy Learning via Action Diffusion, Robotics 2023
>
> [3] Crossway Diffusion: Improving Diffusion-based Visuomotor Policy via Self-supervised Learning, ArXiv 2024
>
> [4] PONI: potential functions for objectgoal navigation with
> interaction-free learning, CVPR 2022
>
> [5] Habitat-Web: Learning Embodied Object-Search Strategies from Human Demonstrations at Scale, CVPR 2022
>
> [6] Peanut: Predicting and navigating to unseen targets, ICCV 2023
>
> [7] Imagine Before Go: Self-Supervised Generative Map for Object Goal Navigation, CVPR 2024

---

> ### Author Response · Authors · 2024-08-12
>
> Thanks to the reviewer for the concerns and suggestions for this work. We hope our response has resolved the confusion and questions in the review. If there are any questions or further comments, please let us know and we will try our best to answer them!

---

> > ### Comment · Reviewer_DH2d · 2024-08-12
> >
> > I acknowledge the authors' comprehensive response, and appreciate the quality of the rebuttal and addition of valuable elements: - real world experiments
> > - clarification of strength of the method for exploration (which I believe warrants a clear discussion in the paper)
> >
> > I have reviewed my score to represent the solidity of the work, although I maintain that novelty is borderline limited.
> > I have no further questions.

---

### Official Review · Reviewer_JCTG · 2024-07-13

**Soundness:** 3
**Presentation:** 3
**Contribution:** 2
**Rating:** 5
**Confidence:** 3

**Summary:**

The paper "Trajectory Diffusion for ObjectGoal Navigation" introduces a novel method called "trajectory diffusion" for the task of ObjectGoal Navigation (ObjectNav), where an agent is required to navigate to a specified object in an unseen environment based on visual observations. The existing methods for ObjectNav often rely on single-step planning, leading to a lack of temporal consistency. The proposed approach leverages diffusion models to learn the distribution of trajectory sequences conditioned on the current observation and the goal. By training with Diffusion Denoising Probabilistic Models (DDPMs) and using optimal trajectory segments, the model can generate a coherent sequence of future trajectories for the agent. The paper demonstrates significant improvements in navigation accuracy and efficiency using the Gibson and MP3D datasets, showcasing the effectiveness of trajectory diffusion in guiding agents in real-world navigation tasks.

**Strengths:**

1. The paper is well-written and clearly positions itself in the literature, highlighting the use of diffusion models to generate a sequence of waypoints for ObjectGoal Navigation.
2. The method is innovative, leveraging diffusion models to enhance temporal consistency in navigation, which addresses a common issue in existing approaches.
3. The evaluations are thorough, with the use of datasets like Gibson and MP3D demonstrating the effectiveness of the approach. The visualizations provided are also helpful in understanding the results.

**Weaknesses:**

1. The paper does not compare its approach with other methods that might use a sequence of waypoints for navigation, leaving a gap in understanding the uniqueness or superiority of using diffusion models for this purpose.
2. The necessity of using diffusion models to predict waypoints is not well justified. There is no comparison with simpler models, such as a standard decoder that outputs a sequence of waypoints, to establish the added value of the diffusion approach.
3. The study might be better suited for a robotics-focused conference rather than NeurIPS, as it leans more towards robot learning than core machine learning innovations.
4. The differences in performance across various hyperparameter settings in the ablation study are relatively small, suggesting that the problem may not be as challenging as presented. For example, varying the length of generated trajectories from 8 to 32 only changes performance by about 3%.
5. There is no exploration of the performance with minimal trajectory lengths, such as 4 or even 1, which could provide insights into the importance of multi-step planning versus single-step planning.

**Questions:**

1. Are there other papers in the literature that use a sequence of waypoints for ObjectGoal Navigation, and how does this method compare?
2. How necessary are diffusion models for predicting waypoints? Could simpler methods like a standard decoder suffice?
3. How would the performance change if the trajectory length were reduced further, possibly to the point of a single-step planner?

---

> ### Author Rebuttal · Authors · 2024-08-07
>
> We appreciate the detailed questions and address them in the following lines.
> ### Comparison with other methods of sequence planning for ObjectNav.
> We choose Habitat-web [1] and ENTL [2] for comparison, as they also output sequence predictions for ObjectNav task. The comparison is based on the following aspects:
>
> - **Training Data.** Both Habitat-web and ENTL rely on human demonstration trajectories for training. In contrast, T-Diff uses automatically collected trajectories, which incur lower collection costs.
> - **Model Structure.** T-Diff's predictions are based on a semantic map. However, both Habitat-web and ENTL rely solely on the implicit encoding of the egocentric view from a few adjacent steps for sequence planning. The semantic map geometrically preserves all past observations, whereas the implicit encoding of a few egocentric views leads to a loss of geometric spatial information, thus limiting their performance in complex environments.
> - **Navigation Performance.** The comparison of navigation performance on MP3D is shown in Table 3 of the main text (for better readability, we have copied the results below). The results indicate that T-Diff achieves higher performance compared to other sequence planning methods, especially with a more significant improvement over ENTL.
>
> These comparisons demonstrate the superiority of using T-Diff for sequence planning.
>
> |              | SR(%) ↑ | SPL(%) ↑ |
> |--------------|---------|----------|
> | ENTL         | 17.0    | 5.0      |
> | Habitat-Web  | 35.4    | 10.2     |
> | T-diff (Ours)| 39.6    | 15.2     |
>
> [1] Habitat-Web: Learning Embodied Object-Search Strategies from Human Demonstrations at Scale. CVPR 2022
> [2] ENTL: Embodied Navigation Trajectory Learner. ICCV 2023
>
> ---
> ### Comparison with simpler model for trajectory generation.
> Please refer to Common Q2 in the Overall Response, where we discuss this concern in detail.
>
> ---
> ### The study emphasizes robot learning over machine learning, possibly unsuitable for NeurIPS.
> We focus on the ObjectNav task, which is related to robotics.
> To efficiently complete this task, we learn a conditional distribution $p(\tau|m_t,o)$, i.e., inferring the trajectory $\tau$ from the current position to the likely location of the target $o$ based on partial observations $m_t$ of the current scene.
> Thus, the foundation of our work lies in learning this conditional distribution, which is inherently a machine learning problem.
>
> Moreover, NeurIPS has established a Robotics area in the main track, which is the track we submitted to.
> Recently, several ObjectNav-related works have been accepted by NeurIPS.
> A subset is listed below.
> Therefore, we believe our work is suitable for NeurIPS.
>
> [1] CaMP: Causal Multi-policy Planning for Interactive Navigation in Multi-room Scenes. NeurIPS2023
> [2] ZSON: Zero-Shot Object-Goal Navigation using Multimodal Goal Embeddings. NeurIPS2022
> [3] ProcTHOR: Large-Scale Embodied AI Using Procedural Generation. NeurlPS2022
> [4] Object Goal Navigation using Goal-Oriented Semantic Exploration NeurIPS2020
>
> ---
> ### The difference in performance across trajectory length from 8 to 32 is relatively small (about 3%).
> We analyze why the navigation performance in Fig. 3 is not significantly affected by hyperparameter variations within a certain range (e.g., trajectory length from 8 to 32). Two possible reasons are:
>
> - **Simulator difficulty.** Experiments in Fig. 3 of the main text are conducted on Gibson, which has a smaller average area (221.33 $m^2$) compared to other datasets (e.g., MP3D, 682.68 $m^2$), making navigation easier and resulting in smaller performance fluctuations. After repeating the experiments on MP3D, we observe that different hyperparameters have a greater impact on performance, causing differences of 8.1% in SR and 3.9% in SPL.
> - **Model robustness.** Another factor is that T-Diff essentially achieves its full potential when the trajectory length is set within a reasonable range (neither excessively large nor small). Consequently, the model's performance exhibits relatively low sensitivity to hyperparameter variations within a certain range.
>
> ---
> ### Performance with minimal trajectory lengths (such as 4 or even 1) of T-Diff.
> We address this concern in the Overall Response. Please refer to Common Q1 for more details.

---

> ### Author Response · Authors · 2024-08-12
>
> Thanks to the reviewer for the concerns and suggestions for this work. We hope our response has resolved the confusion and questions in the review. If there are any questions or further comments, please let us know and we will try our best to answer them!

---

> ### Comment · Area_Chair_QtSE · 2024-08-13
> **Required Action: Please Respond to the Author Rebuttal**
>
> Dear Reviewer JCTG,
>
>
> As the Area Chair for NeurIPS 2024, I am writing to kindly request your attention to the authors' rebuttal for the paper you reviewed.
>
> The authors have provided additional information and clarifications in response to the concerns raised in your initial review. Your insights and expertise are invaluable to our decision-making process, and we would greatly appreciate your assessment of whether the authors' rebuttal adequately addresses your questions or concerns.
>
> Please review the rebuttal and provide feedback. Your continued engagement ensures a fair and thorough review process.
>
> Thank you for your time and dedication to NeurIPS 2024.
>
>
> Best regards,
>
> Area Chair, NeurIPS 2024

---

> ### Comment · Reviewer_JCTG · 2024-08-14
> **Reply to author response**
>
> I appreciate the authors' response to my review, it is helpful for better understanding the paper. I'd like to keep my score as I remain my evaluation to the paper.

---

### Official Review · Reviewer_ria9 · 2024-07-21

**Soundness:** 3
**Presentation:** 3
**Contribution:** 3
**Rating:** 6
**Confidence:** 4

**Summary:**

This paper presents a novel, diffusion-based modular sequential planning algorithm for image-based goal-conditioned ObjectNav tasks. Concretely, the method performs supervised training to solve the following task: given a partial map constructed from past image observations and a user-specified object, predict a finite sequence of future 2D waypoints that would lead the object.

The method's main novelty is a diffusion-based conditional sequence generator. Specifically, the condition denoted $s_t$ is a learned embedding of the incomplete map and the target object. The whole training pipeline starts with data preparation, where ground truth paths are used to generate the incomplete map incrementally, serving as the diffusion model conditions.

The authors perform extensive empirical studies in the Gibson and MP3D simulated environment. The results show that the proposed method is more generalizable across domains than the existing approach. Moreover, it outperforms a collection of end-to-end and modular algorithms, some of which are trained on more data.

**Strengths:**

- The overall presentation of the work is clean and easy to follow.
- The specific learning problem, source of data, and training method are covered in great detail.
- The framework figure is precise and helps the understanding of the approach.
- Using a conditional diffusion model to predict the waypoint sequences makes sense.
- I'm unfamiliar with relevant literature, but the presented empirical results are strong in terms of both in-domain validation results and cross-domain generalization.

**Weaknesses:**

- I find myself scrolling back and forth to read the figures. Maybe their placements could be adjusted.
- The method uses a heuristic $k_g$ to select the one point within the sequence as guidance. While this design seems to work as is, it would be nice if some post-processing on the sequence could be used to pick the point more smartly.
- Additional studies could make the design choices more convincing:
    * If only one point is eventually used as guidance, what would a model perform if trained to directly predict the $k_g$-th point?
    * How would simpler conditional generation models work, for example, directly predicting the sequence or learning a latent variable model?

**Questions:**

- For the hyper-parameter search experiment in Figure 3, how are the other variables chosen when experimenting with one specific parameter?
- The method seems able to plan correct sequences with a very limited initial map. Does this indicate the model learns some inductive bias on where objects are in the rooms? Are there examples where the agent makes a wrong prediction and has to back out?

**Limitations:**

The authors cover the limitations and broader impacts on pages 14 and 15.

---

> ### Author Rebuttal · Authors · 2024-08-07
>
> Thanks to the reviewer for the appreciation and suggestions for our work. We address the concerns in the following lines.
> ### (W1) Figure placement.
> Thanks for the valuable suggestion. We will adjust the figure placement in the final version to enhance readability.
>
> ---
> ### (W2) Post-processing on the sequence for picking the waypoint more smartly.
> Thanks for the valuable suggestion. The parameter $k_g$ is important for selecting the appropriate waypoint on the generated trajectory.
> Previously, point selection was determined through parameter tuning experiments on the validation set.
> As the reviewer suggests, we add post-processing on the generated trajectory to autonomously select the optimal waypoint.
>
> We build an additional trajectory scoring model that takes the generated trajectory, the embedding of semantic map and target as inputs, and outputs a score vector of the same length as the input trajectory.
> Based on collected trajectory segments from training rooms, we calculate the distance of each points on the trajectory from the target location, and construct a one-hot vector with the point closest to the target set to 1 as the training ground truth. The scoring model is trained with Cross-Entropy loss.
>
> By incorporating the score model, the performance of the enhanced T-Diff is shown in the table below. Without hyper-parameter tuning, it achieves comparable or slightly better performance compared to the original version.
> We will include this enhanced version in our final version to further improve the proposed T-Diff.
> |                         | SR(%) ↑  | SPL(%) ↑ | DTS(m) ↓ |
> |-------------------------|----------|----------|----------|
> | T-diff                  | 79.6     | 44.9     | 1.00     |
> | T-diff + s(τ,m_t,o)     | 79.8     | 45.1     | 1.00     |
>
> ---
> ### (W3-1) What would a model perform if trained to directly predict the $k_{g}$-th waypoint?
> We appreciate your concern. This topic is addressed in the Overall Response above, under Common Q1.
>
> ### (W3-2) How would simpler conditional generation models work?
> We discuss this concern in the Overall Response, please refer to Common Q2.
>
> ---
> ### (Q1) How are the other variables chosen when experimenting with one specific parameter in Fig. 3?
> When experimenting with one specific parameter, the values of other variables are held constant. For example, in Fig. 3(c) of the main text, which tests the impact of different $k_g$ values on navigation performance, the length of the trajectory is fixed at $k=32$ and the max de-noising step is set to $\tau=100$.
>
> ---
> ### (Q2-1) Does this indicate the model learns some inductive bias on where objects are in the rooms?
> Yes, T-Diff learns some inductive bias regarding the object layout, which is an expected outcome.
> In the ObjectNav task, agents need prior knowledge about object layout to infer the likely location of the target for more efficient navigation.
> For example, if the target is a sofa, the agent should learn to first plan a trajectory to the living room to explore if the target is there.
> This inductive bias provides a form of prior knowledge to improve navigation efficiency.
>
> However, as the reviewer concerns, this inductive bias can hinder navigation efficiency if it differs from the current room layout.
> Therefore, we control the length of sequence planning (i.e., trajectory length).
> As shown in Fig. 3(a) of the main text, excessively long sequence planning harms navigation performance, while controlling it within a reasonable range ensures that the planning results can be updated in time based on new observations. Within this reasonable range, such inductive bias can improve navigation performance.
> ### (Q2-2) Are there examples where the agent makes a wrong prediction and has to back out?
> We provide visualizations of instances where T-Diff makes incorrect predictions, leading to exploration in the wrong direction, as shown in RFig.2 in the supplementary PDF in the rebuttal.
> The results show that in the initial steps of navigation, due to limited observations of the environment, T-Diff plans an incorrect direction.
> However, since the generated trajectory is controlled to be of a reasonable length and the frequency of new trajectory generation is appropriately set, the agent can quickly change direction (see the orange circle), mitigating the impact of a single-step incorrect prediction by T-Diff.

---

> > ### Comment · Reviewer_ria9 · 2024-08-13
> >
> > I appreciate the authors for addressing my concerns in the original review!
> >
> > The additional experiment results are convincing.
> > - The scoring model seems to help compared to a fixed heuristic step number.
> > - Predicting a trajectory is shown to outperform single-point predictions.
> > - The decoding process is non-trivial, and the diffusion process excels.
> >
> > The additional visualizations in the rebuttal are nice to have and answer my questions.
> >
> > Overall, this work is solid, and I will keep my good rating.

---

> ### Comment · Area_Chair_QtSE · 2024-08-13
> **Required Action: Please Respond to the Author Rebuttal**
>
> Dear Reviewer ria9,
>
>
> As the Area Chair for NeurIPS 2024, I am writing to kindly request your attention to the authors' rebuttal for the paper you reviewed.
>
> The authors have provided additional information and clarifications in response to the concerns raised in your initial review. Your insights and expertise are invaluable to our decision-making process, and we would greatly appreciate your assessment of whether the authors' rebuttal adequately addresses your questions or concerns.
>
> Please review the rebuttal and provide feedback. Your continued engagement ensures a fair and thorough review process.
>
> Thank you for your time and dedication to NeurIPS 2024.
>
>
> Best regards,
>
> Area Chair, NeurIPS 2024

---

### Author Rebuttal · Authors · 2024-08-07

# Overall Response
We thank all reviewers for their valuable and insightful feedback. We appreciate the supportive comments regarding our well-written presentation (ria9, JCTG, sqUP), novel approach (JCTG, sqUP), precise figures (ria9, k5Ci), sound motivation (ria9), robust performance (ria9, k5Ci, sqUP), and comprehensive evaluations (JCTG, DH2d).
Several common questions have been raised, which we address below.

### Common Q1 (@ria9 @JCTG) Using minimal trajectory lengths for T-Diff training.
As suggested by the reviewers, we add an ablation study using minimal trajectory lengths (e.g., 1 and 4) for training T-Diff, as shown in RTab.1 in the supplementary PDF in the rebuttal. Note that when the length is set to 1, the prediction of T-Diff is a single waypoint.

The results indicate that when the length is set to 1, T-Diff's performance is influenced by the choice of ground truth point (i.e., the **i-th** point from current position on the optimal trajectory). The performance with shorter lengths (1 or 4) is lower compared to longer lengths (32).

We hypothesize that predicting a sequence of trajectories, as opposed to a single point, allows each predicted point to receive contextual information from neighboring points. This helps correct and smooth out prediction errors of individual points, reducing the sensitivity of the results to single-point errors. Consequently, this ensures more stable predictions and enhances overall accuracy of trajectory prediction.
This finding further supports our motivation for using sequence planning.

---
### Common Q2 (@ria9 @JCTG) Comparison with simpler model for trajectory generation.
We consider the following simple decoder to learn the trajectory generation $P(\tau|m_t, o)$ as a comparison.
It adopts a similar Transformer-based architecture to T-Diff, with comparable parameters and the same conditional inputs.
However, unlike T-Diff, which is trained using DDPM, this competitor is trained with MSE loss. The results are shown in the table below.

The results indicate that directly learning $P(\tau|m_t, o)$ through supervised training yields poor performance. Our analysis suggests that since both $m_t$ and $\tau$ are high-dimensional, the target distribution $P(\tau|m_t, o)$ is also high-dimensional. Given the limited number of training rooms (less than 100), $P(\tau|m_t, o)$ is sparse and difficult to learn directly.
In contrast, the diffusion model (DDPM), through its diffusion and denoising process, gradually simplifies complex distribution into multiple simpler distributions.
This allows for better learning of $P(\tau|m_t, o)$ distribution.
Therefore, our experiments and analysis confirm the necessity of using diffusion models for learning trajectory generation.
||MSE ↓|SR(%) ↑|SPL(%) ↑|DTS(m) ↓|
|-|-|-|-|-|
|Simple decoder|0.6541|59.2|33.5|2.05|
|T-diff|0.0357|79.6|44.9|1.00|

> Note that MSE measures the quality of generated trajectories, while SR, SPL, and DTS indicate navigation performance.

---
##### Common Q3(@DH2d @sqUP) Concerns about map size.
We conduct experiments with T-Diff using various map sizes. Results are presented in RFig.3 of the supplementary PDF. Our findings demonstrate that T-Diff is compatible and performs well with different map sizes (except when excessively small). The results indicate that as map size increases, both information granularity and performance improve, albeit with increased computational complexity. Performance plateaus beyond a size of 300, while complexity continues to rise. To balance performance and computational cost, we opt for a 224×224 map size. However, when T-Diff requires adaptation to larger-scale scenarios, map size can be scaled up and readily integrated (addressing **sqUP**'s concerns about small map sizes limiting T-Diff's performance in large-scale environments). Additionally, we observe that performance degrades significantly when the map size falls below 150 (addressing **DH2d**'s inquiry about the point at which map size constrains performance).

---
### Common Q4 (@DH2d @sqUP) Real world experiments.
To validate T-diff in real-world environments, we provide additional evaluation results and navigation visualizations in real world, as shown in the supplementary PDF in the rebuttal.
We set up a 140 $m^2$ space, utilizing movable walls to create 3 scenes with different layouts. Each scene is divided into several rooms and furnished with common furniture and objects. The space contains a total of 35 object categories, from which we select 7 object types for our object navigation experiments. The experiments are deployed on a Locobot-wx250s. For each object type, we conduct trials from 10 different starting positions in each of the 3 scenes, and calculate the SR metrics.
As shown in RTab.2, our method achieves a higher success rate compared to the baseline (PONI[1]). Additionally, the visualizations in RFig.1 show that the generated trajectories align well with actual target positions, demonstrating T-diff's robustness in real-world scenarios.


---
### Common Q5 (@DH2d @sqUP) Misunderstanding regarding the result of row 1 in Tab. 1 in the main text.
In Tab. 1 of the main text, rows 2-4 represent different variants of T-Diff (i.e., sequence planner with geometric memory).
The comparison in row 1 uses an enhanced FBE method (proposed by PONI[1]) combined with a local policy (i.e., single-step planner with geometric memory).
The 'X' marks under 'visual' and 'goal' condition in row 1 indicate that T-Diff is not used, but this alternative still employs semantic map and goal for navigation.

To improve clarity, we revise this table, as shown in RTab.3 in the supplementary PDF in the rebuttal. RTab.3 provides clearer explanations and includes an additional row (row 0) to address the reviewer's concern about the case where navigation process does not use any image or goal.

---
### Reference
[1] PONI: potential functions for objectgoal navigation with interaction-free learning, CVPR 2022

---

### Decision · Program_Chairs · 2024-09-25

**Decision:**

Accept (poster)

**Comment:**

The paper "Trajectory Diffusion for ObjectGoal Navigation" presents a novel approach for object goal navigation using a trajectory diffusion model called T-Diff. The proposed method leverages a conditional diffusion model to generate trajectories based on a semantic map and target object, guiding the agent toward potential target locations in unseen environments. The reviewers agree that the paper makes a contribution to the field of embodied AI and recommend its acceptance. The strengths of the paper lie in its innovative use of a diffusion model for trajectory generation in the context of object goal navigation. By conditioning the diffusion process on a semantic map and target object, T-Diff can generate temporally consistent and interpretable trajectories that guide the agent towards potential target locations in unobserved areas. This approach addresses the limitations of existing end-to-end and modular methods, which often suffer from temporal inconsistencies and inefficient exploration.

The reviewers raised several important questions and concerns, which the authors addressed in their rebuttal. They provided additional experiments and analyses to support their claims, including the effectiveness of using optimal trajectories for training, the impact of map size on performance, and the comparison with simpler models for trajectory generation. The authors also clarified the differences between T-Diff and related works, such as modular methods and zero-shot navigation approaches. The authors addressed the concerns regarding the temporal consistency of end-to-end and modular methods, and provided evidence that T-Diff achieves more consistent planning by considering past trajectories and current agent pose. They also explained the advantages of sequence planning over single-step prediction, highlighting the importance of balancing the sequence length to ensure timely corrections. While some reviewers raised concerns about the small map size potentially limiting T-Diff's generalization to large-scale environments, the authors demonstrated that their method is compatible with different map sizes and can be scaled up for larger environments. They also provided additional real-world experiments to validate the effectiveness of T-Diff in practical scenarios.

The AC also considers this work to be both interesting and important, as it introduces a novel approach for object goal navigation that addresses the limitations of existing methods. The use of trajectory diffusion for generating temporally consistent and interpretable trajectories based on semantic maps and target objects is a significant contribution to the field. The experiments and analyses provided by the authors further demonstrate the effectiveness and robustness of their approach. The reviewers' concerns have been adequately addressed in the rebuttal, and the authors have shown a commitment to improving the paper based on the feedback received.

Based on the above reasons, the AC recommends accepting this paper for publication.